# The immune landscape of human thymic epithelial tumors

Zhongwei Xin [1,2,5], Mingjie Lin[1,2,5], Zhixing Hao[1,2,5], Di Chen[2,3], Yongyuan Chen[1,2], Xiaoke Chen[1,2], Xia Xu[4], Jinfan Li[4], Dang Wu[2,3], Ying Chai [1] ✉ & Pin Wu [1,2] ✉

Human thymic epithelial tumors (TET) are common malignancies in the anterior mediastinum with limited biological understanding. Here we show, by single cell analysis of the immune landscape, that the developmental pattern of intra-tumoral T-cells identify three types within TETs. We characterize the developmental alterations and TCR repertoires of tumor-infiltrating T cells in the context of the distinguishing epithelial tumor cell types. We demonstrate that a subset of tumor cells, featuring medullary thymic epithelial cell (TEC) phenotype and marked by *KRT14/GNB3* expression, accumulate in type 1 TETs, while T-cell positive selection is inhibited. Type 2 TETs are dominated by *CCL25*+ cortical TEC-like cells that appear to promote T-cell positive selection. Interestingly, the *CHI3L1*+ medullary TEC-like cells that are the characteristic feature of type 3 TETs don't seem to support T-cell development, however, they may induce a tissue-resident CD8+ T cell response. In summary, our work suggests that the molecular subtype of epithelial tumour cells in TETs determine their tumour immune microenvironment, thus *GNB3* and *CHI3L1* might predict the immunological behavior and hence prognosis of these tumours.

Thymic epithelial tumors (TET), including thymomas and thymic carcinomas, are common primary tumors in the human anterior mediastinum[1,2]. Owing to the lack of relevant cell lines and animal models[3–6], the understanding of the etiology and biology of TETs is far from adequate. All types of TETs are considered potentially malignant in the latest clinical guidelines[7] because previously defined benign tumors also exhibit aggressive malignant behavior clinically[8]. This suggests that further understand the biological characteristics of these heterogeneous malignancies is needed.

Generally, TETs are considered to be transformed from thymic epithelial cells (TEC), which play a crucial role in T-cell development in mammals[9]. There are two major subtypes of TECs that may be distinguished according to the epithelial origins of human TETs, cortical TECs (cTEC) and medullary TECs (mTEC), which are mainly located in the cortex and medulla structures of the thymus, respectively[10]. It is well demonstrated that cTECs are essential for the positive selection of T cells during T-cell development in the thymus, while mTECs are mainly involved in negative selection to establish central self-tolerance[11,12]. Clinical studies have shown that patients with type B thymoma, which may be derived from mTECs, have a higher incidence of autoimmune diseases than patients with type A and type C thymoma[13,14]. This evidence suggests that the impact of tumor cells on T-cell development in human TETs may be different. Recently, emerging molecular classifications have provided insight into the stratification of prognosis and treatment for patients with TETs at the molecular level[15,16].

Here we conduct a comprehensive study using mass cytometry, single-cell sequencing, TCR repertoires, histological analysis, FCM

[1]Department of Thoracic Surgery, The Second Affiliated Hospital, Zhejiang University School of Medicine, Zhejiang University, Hangzhou 310009, China. [2]Key Laboratory of Tumor Microenvironment and Immune Therapy of Zhejiang Province, The Second Affiliated Hospital, Zhejiang University School of Medicine, Zhejiang University, Hangzhou 310009, China. [3]Department of Oncology Radiotherapy, The Second Affiliated Hospital, Zhejiang University School of Medicine, Zhejiang University, Hangzhou 310009, China. [4]Department of Pathology, The Second Affiliated Hospital, Zhejiang University School of Medicine, Zhejiang University, Hangzhou 310009, China. [5]These authors contributed equally: Zhongwei Xin, Mingjie Lin, Zhixing Hao. ✉e-mail: chaiy@zju.edu.cn; pinwu@zju.edu.cn

detection and immunofluorescence testing to decode the epithelial origins of human TETs and their impact on immune cell composition. Based on the phenotype of tumor cells, the immune landscape of tumors, intra-tumoral T-cell development pattern and TCR repertoire, we identify three main types of human TETs and uncover the potential mechanisms that lead to the divergent T-cell development in each tumor type.

## Results

### The immune landscape of human TETs is heterogeneous

Using multiple approaches, we performed a comprehensive study of the immune landscape of human TETs (Fig. 1a). First, we established a panel of more than 40 markers to detect the cell composition of the human thymus and TETs by CyTOF and found a remarkable difference between the tumor and thymus (Supplementary Fig. 1a, Supplementary Table 1 and Table 2). An apparent discrepancy in the immune landscape also existed among samples of human TETs (Supplementary Fig. 1a), which canonical classifications were not fully aligned (Fig. 1b and Supplementary Fig. 1a). According to the similarity of their cellular composition, we grouped the samples of human TETs into three types (type 1, 2 and 3) (Fig. 1b and Supplementary Fig. 1a). Interestingly, we found that the variance in the immune landscape among the types according to our alternative classification was not fully aligned with the WHO classification and Masaoka stage (Fig. 1b and Supplementary Fig. 1a). The reliability of our alternative classification was further validated by multiple clustering schemes and histological analysis (Supplementary Fig. 1b, c and d). Statistical analysis showed that the quantity of macro-cluster cells was significantly different between groups according to our alternative classification (Fig. 1c, d and Supplementary Fig. 1a, e, f). In addition, each type of tumor in our potential alternative classification contained heterogeneous samples according to the WHO subtype and Masaoka stage (Fig. 1e, f).

Through cell annotation, we found that $CD45^+$ immune cells, but not $EPCAM^+$ tumor cells, constituted the dominant cell subset in TET tumors (Fig. 1g and Supplementary Fig. 1g). In addition, $CD45^+$ immune cells, but not epithelial cells, constituted the major quantitative variate among samples grouped by our alternative classification, WHO classification and Masaoka stage (Fig. 1b, g and Supplementary Fig. 1h). Further cell annotation showed that both TETs and the normal thymus comprised major immune lineages (Fig. 1h, i and Supplementary Fig. 2a), with abundant $CD3^-CD4^+CD8^+$ and $CD3^+$ T cells (Supplementary Fig. 2b). Intergroup analysis showed obvious differences in the immune cell composition of tumors, especially $CD3^-CD4^+CD8^+$ and $CD3^+$ T lymphocytes, which are involved in intrathymic T-cell development (Fig. 1j, k). In particular, the proportion of $CD3^-CD4^+CD8^+$ lymphocytes was the highest in type 1 and lowest in type 3 TETs (Fig. 1k). Moreover, from type 1 to type 3 TETs, there was an obvious tendency toward a decreased abundance of $CD3^-CD4^+CD8^+$ lymphocytes in tumors (Fig. 1k). However, the proportions of $CD3^+$ T cells in tissues of both type 2 and type 3 TETs were higher than those in type 1 TETs (Supplementary Fig. 2c). In contrast, the proportions of B cells, dendritic cells (DC), granulocytes and natural killer cells (NK) in tumors was lowest in type 1 and highest in type 3 TETs, with a tendency toward a gradual increase in abundance from type 1 to type 3 TETs (Fig. 1k and Supplementary Fig. 2c). Monocytes were abundant in tumors of type 3 (Supplementary Fig. 2c). It is interesting that except for $CD3^-CD4^+CD8^+$ lymphocytes, other immune cells related to the peripheral immune response in tumors were more abundant in type 3 TETs (Fig. 1k and Supplementary Fig. 2c). The discrepant abundance of $CD3^-CD4^+CD8^+$ lymphocytes and $CD3^+$ T-cell tumor types was further validated by flow cytometry (Fig. 1l). Importantly, compared with the WHO classification and Masaoka staging system, our system is focusing on the differences in immune cell subsets in tumors in a comprehensive manner (Supplementary Fig. 3).

### Intratumoral T-cell development pattern of human TETs

The varying proportions of $CD3^-CD4^+CD8^+$ and $CD3^+$ T lymphocytes among tumors suggested that there was a different T-cell development pattern in each TET type. To further investigate the change in T-cell development in TETs, we used well-accepted markers, which are highly correlated with T-cell development in the thymus[17], to annotate the subsets of $CD3^+$ T cells in the tumor and normal thymus (Fig. 2a and Supplementary Figs. 4a, b). Major subsets of T cells involved in thymic T-cell development were represented[12], including double-negative (DN, $CD4^-CD8^-$), double-positive (DP, $CD4^+CD8^+$), $CD4^+$ single-positive ($CD4^+$ SP), $CD8^+$ single-positive ($CD8^+$ SP), and $FOXP3^+$ regulatory T (Treg) cells (Fig. 2a). Phenotypic analysis showed that the T-cell subsets in thymus tissues and tumors that we defined were representative (Fig. 2b). It is important that the differences in T-cell composition among tumors of different subtypes were well defined by our immuno-based alternative classification, while the WHO classification or Masaoka staging system was more focused on the histopathology and morphology of tumor (Fig. 2c–f). In our alternative classification, immature DP cells dominated in type 1 TETs, while mature SP cells dominated in tumors of type 2 and type 3 (Fig. 2f, g). The proportion of DN and DP cells was the highest in type 1 tumors and lowest in type 3 tumors (Fig. 2g and Supplementary Fig. 4c). Interestingly, the proportion of DN and DP cells in TETs, especially DP cells, showed a tendency to decrease from type 1 to type 3 TETs (Fig. 2g and Supplementary Fig. 4c). In contrast, the proportion of $CD4^+$ and $CD8^+$ T cells was lowest in tumors of type 1 and highest in tumors of type 3 (Fig. 2g). In addition, the proportions of $CD4^+$ and $CD8^+$ T cells in tumors showed an increasing trend from type 1 to type 3 TETs, which was contrary to the pattern observed for DN and DP cells (Fig. 2g). Flow cytometry detection further confirmed the different proportions of DN, DP and SP cells in each TET type (Fig. 2h). Furthermore, immunofluorescence staining showed that DP and SP cells were located in the cortex- and medulla-like structure in type 2 TETs, respectively; however, the typical cortex- and medulla-like structures were not observed in type 1 and type 3 TETs (Fig. 2i).

Regarding mature T cells, the proportions of naïve SP cells and memory SP cells were lowest in type 1 TETs and highest in type 2 TETs (Fig. 2g and Supplementary Fig. 4c). Interestingly, tissue-resident $CD4^+$ T ($CD4^+$ $T_{RM}$) cells, tissue-resident $CD8^+$ T ($CD8^+$ $T_{RM}$) cells and regulatory T cells (Treg), which are typically present in peripheral tissues[18], showed an increasing tendency in type 3 TETs (Supplementary Fig. 4c). In addition, we found that the ratio of $CD4^+$ to $CD8^+$ T cells was lowest in tumors of type 1 (Supplementary Fig. 4c). Meanwhile, the differences in mature T-cell composition among WHO classification and Masaoka staging system based on the histopathology and morphology were similar to that in our alternative classification (Supplementary Fig. 5). Taken together, these findings uncovered a unique pattern of T-cell development in each type of tumor according to our alternative classification.

### Dysfunction of T-cell development in human TETs

To validate the differences in immune cell composition among tumor types, we performed scRNA-seq and TCRαβ profiling on six paired tumor samples, which validated the immunosubtypes from our alternative classification (2 samples each type) (Fig. 1a). Data from a normal thymus sample published by Jong-Eun Park et al. were also reanalyzed as a normal control[17]. A total of 52,788 cells from tumors and 2,845 cells from the normal thymus were included (Supplementary Table 3). Twenty-six cell clusters were identified (Supplementary Fig. 6a). Using well-known lineage markers, we further annotated the cell clusters into nine major cell types (Supplementary Fig. 6b–e and Supplementary Data 1). Consistent with the results of CyTOF, the cellular atlas showed that T cells accounted for the major differences in cell types between tumor and normal tissue as well as among types of TETs (Supplementary Fig. 6f and g). In addition, the proportions of B cells and

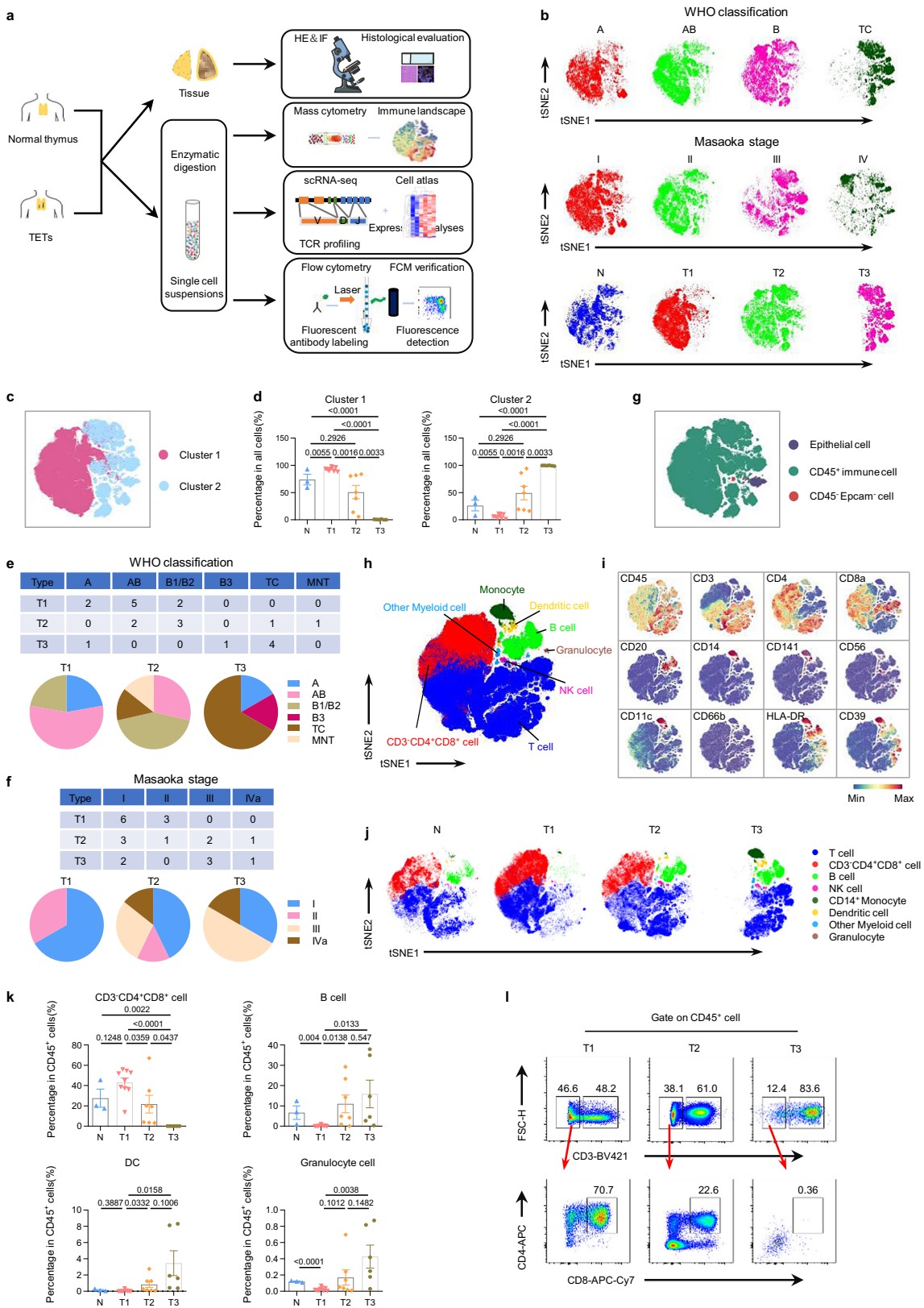

monocytes showed a similar tendency to vary among tumor types as the observed in the CyTOF results (Supplementary Fig. 6h).

To further investigate the differences in T-cell development patterns among tumor types, T cells were reclustered into 18 cell clusters (Supplementary Fig. 7a). Then, the cell clusters were further annotated (Fig. 3a), according to the markers related to T-cell development

(Fig. 3b, Supplementary Fig. 7b and Supplementary Data 2). The characteristic molecules of tumor-infiltrating T cells detected by CyTOF were validated by scRNA-seq (Fig. 2b and Supplementary Fig. 7c). In addition, two immature T-cell subsets in tumors (proliferating (P) DP cells and quiescent (Q) DP cells) were identified by using *MKI67* and *CDK1* according to a previous study[17] (Supplementary

**Fig. 1 | Cellular composition of the human thymus and TETs.** (**a**) Schematic diagram of the overall study design and workflow. (**b**) t-SNE plots of total cells from human TETs (*n* = 22) and the normal thymus (*n* = 3) by CyTOF using a Phenograph clustering scheme. Samples were grouped according to WHO histological subtype, Masaoka stage or our alternative classification, and each color represents an independent group or stage (N, normal human thymus; T1, type 1; T2, type 2; T3, type 3; the abbreviations below are consistent with these definitions). (**c**) t-SNE plot of total cells from all samples (*n* = 25) by CyTOF using a Phenograph clustering scheme, colored by cell cluster. (**d**) Bar plots showing the frequencies of cluster 1 and cluster 2 subsets among the four groups of samples (*n* = 3, 9, 7 and 6 for N, T1, T2 and T3, respectively. Data are presented as the mean ± s.e.m. *P* values in the figure were determined by an unpaired two-tailed Student's t test). (**e-f**) Sheet and pie charts summarizing samples in each alternative classification by WHO histological subtype (**e**) or Masaoka stage (**f**). Each color represents a WHO histological subtype or Masaoka stage. (**g**) Same t-SNE plot as (**c**), colored by cell type. (**h**) t-SNE plot of immune cells (EPCAM-CD45 + ) from all samples obtained by CyTOF using a Phenograph clustering scheme, colored by cell type. (**i**) t-SNE analysis of immune cells from all samples colored by the relative expression of CyTOF markers (CD45, CD3, CD4, CD8a, CD20, CD14, CD141, CD56, CD11c, CD66b, HLA-DR and CD39). (**j**) Same t-SNE plots of immune cells as (**h**) from samples of each group (N, T1, T2, T3), colored by cell type. (**k**) Bar plots showing the frequencies of the main immune cell subsets among the four groups of samples (*n* = 3, 9, 7 and 6 for N, T1, T2 and T3, respectively. Data are presented as the mean ± s.e.m. *P* values in the figure were determined by an unpaired two-tailed Student's t test). (**l**) Representative flow cytometric plot of CD3+ T cells and CD3-CD4+CD8+ lymphocytes from TETs of each type. Source data are provided as a Source Data file.

Fig. 7d–f); the result was partly consistent with the findings obtained via CyTOF (Supplementary Fig. 7g). Interestingly, the expression levels of CD3 genes were not distinct among CD3+ DP cells and CD3- DP cells (Supplementary Fig. 7h), which suggested that these immature DP (P) cells corresponded with the CD3- DP cells in CyTOF analysis.

Compared with the normal thymus, we found that early DP cells increased in type 1 TETs but decreased in type 2 TETs and were almost absent in type 3 TETs (Fig. 3c–e and Supplementary Fig. 7i). In contrast, the abundance of mature SP cells was significantly increased in type 2 and type 3 TETs (Fig. 3c–e and Supplementary Fig. 7i). Moreover, the composition of T-cell subsets involved in T-cell development was significantly different among different types of TETs (Fig. 3c–e). Specifically, early DP cells constituted the largest T-cell subset in type 1 TETs but the smallest subset in type 3 TETs (Fig. 3d, e and Supplementary Fig. 7i). In contrast, mature SP cells constituted the largest T-cell subset in type 3 TETs but the smallest T-cell subset in type 1 TETs (Fig. 3d, e and Supplementary Fig. 7i). Interestingly, naïve T cells were more abundant in type 2 TETs than in type 3 TETs, whereas memory T cells were more abundant in type 3 TETs than in type 2 TETs (Fig. 3d, e and Supplementary Fig. 7i).

To further validate the developmental differences among T cells in tumors, we performed trajectory analysis of the previously defined T-cell subpopulations (Fig. 3f). Consistent with previous findings, trajectory analysis showed an obvious difference in T-cell development in each type of TET compared with the normal thymus (Fig. 3g). Through intergroup analysis, we found that T cells in type 1 TETs were enriched in the early stage of development, while T cells in type 2 TETs were enriched in both the early stage and late stages of development (Fig. 3g). Interestingly, almost all T cells were concentrated in the late stage of development in type 3 TETs (Fig. 3g). The gene expression levels of Notch and Wnt signaling components, which regulate the early stages of T-cell development[19,20], were higher in the T cells isolated from type 1 TETs than in those isolated from type 2 and type 3 TETs (Fig. 3h and i). In contrast, IL-7 signaling-related molecules, which could drive intrathymic expansion of positively selected thymocytes prior to their export to the peripheral T-cell pool[21], were expressed at higher levels in T cells isolated from type 2 TETs than in those isolated from the normal thymus and other tumor types (Fig. 3j). These results suggested that the course of T-cell development was restrained in the early stage in type 1, promoted in type 2 and almost abolished in type 3 human TETs.

## Bias of the TCR repertoire and TCR diversity in human TETs

The TCR repertoire and TCR diversity are closely related to intrathymic T-cell development[17]. For TCRβ, we found an obvious bias in VJ gene usage among tumor types of TETs (Fig. 4a), which resembled the VJ gene usage pattern of DP and SP cells (Supplementary Fig. 8a). Consistently, the VJ pairs of T cells were also significantly different among different types of TETs (Fig. 4b) and were consistent with DP and SP cells (Supplementary Fig. 8b). For the TCRα locus, we found a clear association between developmental timing and V-J pairing

(Supplementary Fig. 8b), as previously described[17,22]. During the T-cell developmental stage, recombination of proximal pairs at the TCRα locus was mainly observed in the DP stage, whereas recombination of distal pairs was mainly observed in the SP stage (Supplementary Fig. 8c). This bias in recombination in turn restricted the V-J pairing of TCRα in DP and SP cells (Supplementary Fig. 8d). We also observed that the VJ gene usage and pairing of TCRα in each tumor type were in a similar pattern as DP and SP cells respectively. (Fig. 4c, d). Furthermore, we found that the clonotypes of tumor-infiltrating T cells were strongly associated with the T-cell developmental states defined by scRNA-Seq (Fig. 4e). It is worth noting that clonotypic amplification was mainly observed in mature SP cells but was less frequent in immature DN and DP cells (Fig. 4e). This also indicated that immature T cells (DN and DP cells) had higher clonal diversity than mature T cells (SP cells), which was consistent with the finding of low clonal diversity and high numbers of SP cells in type 3 TETs (Fig. 4f). Together, these findings uncovered the potential effects of the T development differences among human TET types on the TCR repertoire and clonal amplification.

## Epithelial origin decodes T-cell developmental dysfunction in human TETs

Intrathymic T-cell development is tightly orchestrated by heterogeneous TECs[10]. To further investigate the epithelial origin of each type of TET and its potential impact on T-cell development, the subsets of epithelial cells (tumor cells) in the thymus and tumors were further analyzed (Fig. 5a and Supplementary Fig. 9a). A varying epithelial cell composition among human TETs was observed (Fig. 5a). Epithelial cells were most abundant in type 3 TETs, while they were much less abundant in type 1 and type 2 TETs (Fig. 5a, b). Gene expression analysis showed that epithelial cells in type 3 TETs expressed high levels of cancer stem cell (CSC)-related markers (*NOTCH2*, *CD24*, *ALDH1B1* and *PROM1*) (Fig. 5c). At the protein level, the epithelial cells in type 3 TETs expressed higher levels of Ki67, further suggesting malignancy (Supplementary Fig. 9b). In addition, trajectory analysis showed that the epithelial cells in type 1 and type 2 TETs were at discrete developmental stages compared with those in type 3 TETs (Fig. 5d).

To further uncover the heterogeneity of tumor cells among TET subtypes, the epithelial cells in TETs were reclustered into 17 cell clusters (Supplementary Fig. 9c). The cell clusters were annotated into seven subsets based on a combination of known TEC markers (Fig. 5e–g and Supplementary Fig. 9d and Supplementary Data 3). Through gene expression analysis, we found that GNB3+ mTEC-like cells expressed high levels of *OVOL3*, *PLCB2*, and *TRPM5* but low levels of *PLCG2*, *CHAT*, and *DCLK1*, all of which are marker genes of tuft-like mTEC cells in the normal thymus[17,23,24] (Supplementary Fig. 9e). Similarly, *KRT14*+ mTEC-like cells expressed high levels of *CLU*, *FN1*, and *IGFBP5* but not *ITGA6*, *KRT17*, or *SOX4*[17,24] (Supplementary Fig. 9f). Furthermore, we found that *CTSL* and *PRSS16* were mainly expressed in cTEC-like cells and mcTEC-like cells in TETs (Supplementary Fig. 9g). *FEZF2* and *AIRE*, which are responsible for thymic T-cell negative

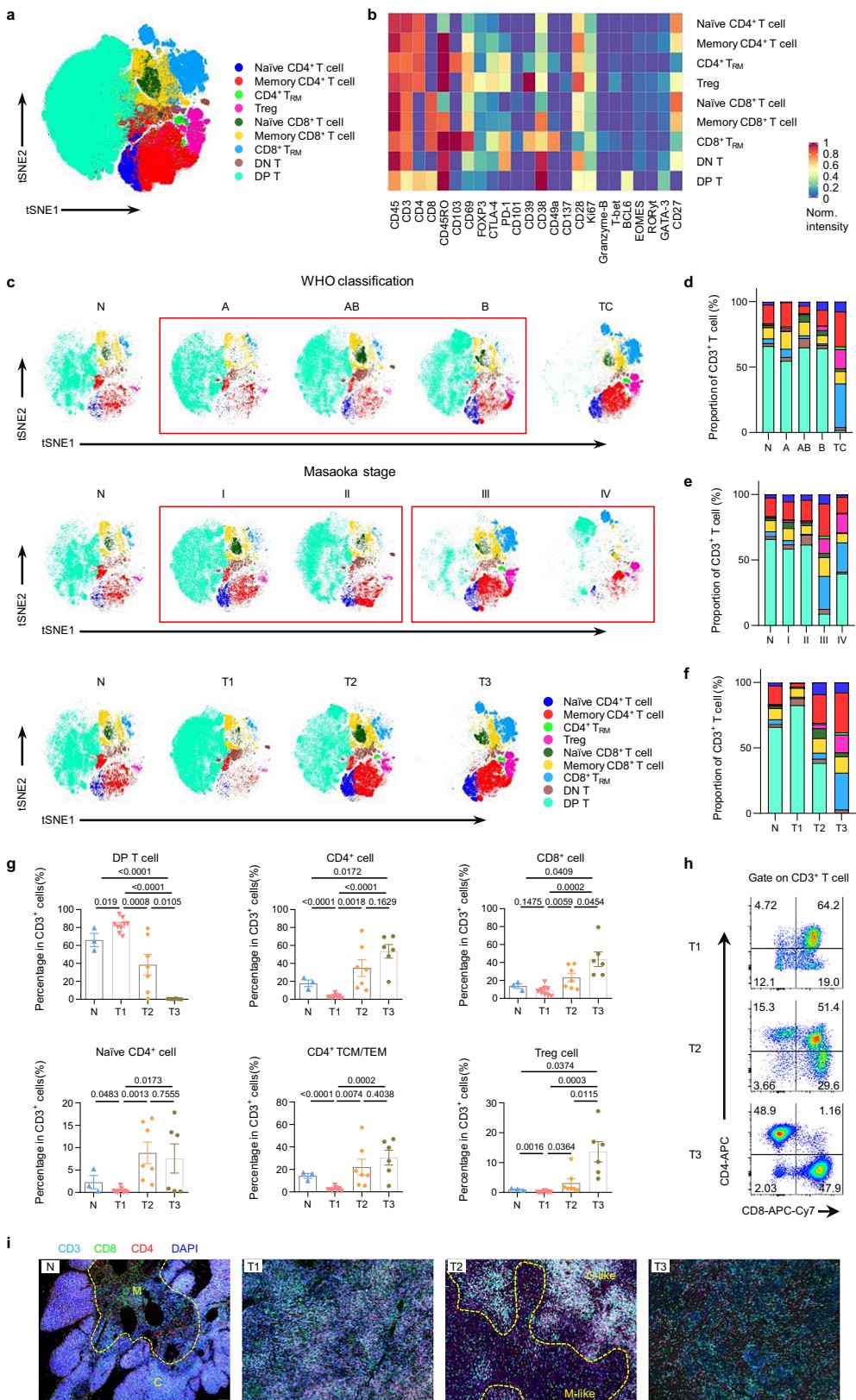

selection, were mainly expressed in *CHI3L1*⁺ mTEC-like cells, *MYOG*⁺ TEC-like cells, *GNB3*⁺ mTEC-like cells and *CHGA*⁺ TEC-like cells (Supplementary Fig. 9g). Intergroup analysis showed that the composition of tumor cell subsets and TEC marker gene expression in different TET subtypes was significantly different (Fig. 5e, h and Supplementary Fig. 10). In addition, our results identified the dominant tumor cell

subsets in each type of TET (Fig. 5e and h). The main tumor cell subsets in each TET type were further confirmed by immunofluorescence (Fig. 5i). Interestingly, mcTEC-like cells present in both type 1 and type 2 tumors exhibited more molecular characteristics of thymic epithelial progenitor cells (TEPCs) than *KRT14*⁺ mTEC-like and *CCL25*⁺ cTEC-like cells[25] (Supplementary Fig. 11a). RNA velocity indicated that *KRT14*⁺ and

**Fig. 2 | T-cell subsets in the thymus and TET subtypes.** (a) t-SNE plot of CD45⁺CD3⁺ T cells from all samples (n = 25), colored by cell subset. (DN, CD3⁺CD4⁻CD8⁻; DP, CD3⁺CD4⁺CD8⁺). (**b**) Heatmap showing relative normalized protein expression for the T-cell subsets from CyTOF, normalized total matrix. (**c**) t-SNE plots of CD45⁺CD3⁺ T cells from human TETs (n = 22) and the normal thymus (n = 3) by CyTOF using the Phenograph clustering scheme. Samples were grouped according to WHO classification, Masaoka stage or our alternative classification, and each color represents a T-cell subset. (**d–f**) Composition of the CD45⁺CD3⁺ T-cell compartment showing the average frequencies of major T-cell subsets for each WHO histological subtype (**d**) (n = 3, 3, 7, 6 and 5 for N, A, AB, B and TC, respectively), Masaoka stage (**e**) (n = 3, 11, 4, 5 and 2 for N, I, II, III and IV, respectively) or our alternative classification (**f**) (n = 3, 9, 7 and 6 for N, T1, T2 and T3,

respectively). Each color represents a T-cell subset, same as in (**c**). (**g**) Bar plots showing the frequencies of the main T-cell subsets among the four groups of samples (n = 3, 9, 7 and 6 for N, T1, T2 and T3, respectively). Data are presented as the mean ± s.e.m. *P* values in the figure were determined by an unpaired two-tailed Student's t test. (DP, CD3⁺CD4⁺CD8⁺). (**h**) Representative flow cytometric plot of CD3⁺CD4⁻CD8⁻ T cells, CD3⁺CD4⁺ T cells, CD3⁺CD8⁺ T cells and CD3⁺CD4⁺CD8⁺ T cells from TETs of three types. (**i**) Representative immunofluorescence (IF) staining images showing CD3 (pale blue), CD8 (green), CD4 (red) and DAPI (nuclei, blue) in the normal human thymus and TET samples (scale bar: 100 μm). (M: Medulla; C: Cortex; M-like: Medulla-like structure; C-like: Cortex-like structure). Experiment was performed in three independent samples for each group with similar results. Source data are provided as a Source Data file.

GNB3 + mTEC-like cells might be differentiated from mcTEC-like cells (Supplementary Fig. 11b). Through pathway analysis, we found that NF-κB signaling in mcTEC-like cells of type 1 TETs was obviously activated, while it was inhibited in type 2 TETs (Supplementary Fig. 11c), which implied that the differentiation of mcTEC-like cells into KRT14 + and GNB3 + mTEC-like cells in type 1 TETs was promoted, while it was suppressed in type 2 TETs[25]. Moreover, *AIRE* expression in type 2 TETs was significantly lower than that in the other two types (Supplementary Fig. 12a and b), which was correlated with the incidence of autoimmune disease[13]. Consistently, we also observed a higher proportion of type 2 TET patients presenting with myasthenia gravis (MG) (Supplementary Fig. 12c). It is interesting that expression of CHRNA1 was only detected in MYOG + TEC-like cells of type 3 TETs, which indicated that the loss of CHRNA1 + MYOG + TEC-like cells may involve in the MG development in type 1 and type 2 TETs[26,27] (Supplementary Fig. 12d, e).

Interestingly, CXCL12 and CCL25, the key chemokines recruiting blood-borne lymphoid progenitor cells into the thymus[25], were almost not expressed in *CHI3L1*⁺ mTEC-like cells (Supplementary Fig. 13a). Then, we focused on the tumor cell and lymphocyte interactions mediated by chemokines and demonstrated that mTEC-like and mcTEC-like tumor cells in TETs could induce DP cell migration for positive selection through *CCL25:CCR9* and *CXCL12:CXCR3* interactions (Supplementary Fig. 13b and Supplementary Data 4). Interestingly, cTEC-like tumor cells also played a role in inducing SP cell migration through the *CCL19:CCR7* interaction (Supplementary Fig. 13c), which differed from cTECs in the normal thymus[28].

Through Kyoto Encyclopedia of Genes and Genomes (KEGG) analysis, we found that the genes highly expressed in DP cells of type 2 TETs were significantly enriched in signaling pathways related to T-cell expansion and positive selection, including oxidative phosphorylation, the NF-κB signaling pathway, Th17-cell differentiation, Th1 and Th2 cell differentiation and the TNF signaling pathway[25,29] (Supplementary Fig. 13d). High expression of genes in apoptosis-related pathways also indicated that more DP cells in type 2 TETs were undergoing positive selection (Supplementary Fig. 13d). Compared with normal thymus tissue, DP cells of type 2 TETs showed high expression of genes enriched in signaling pathways relevant for positive selection (Supplementary Fig. 13e). Moreover, we analyzed the expression of marker genes that indicated the pre- or ongoing state for positive selection of DP cells[29] and found that DP cells in type 1 TETs mainly expressed preselection-related genes, while DP cells in type 2 TETs highly expressed genes related to the ongoing state of positive selection (Supplementary Fig. 13 f). These results indicated that positive selection during T-cell development was promoted in type 2 TETs but blocked in type 1 TETs, which was further supported by the observation that the expression level of HLA-DR in tumor cells in type 1 TETs was lower than that in other TET types (Supplementary Fig. 13 g). Furthermore, we performed a coculture experiment using sorted tumor epithelial cells and DP cells of type 1 and type 2 TETs (Supplementary Fig. 13h-j) and showed that tumor cells could directly promote T-cell development in vitro (Supplementary Fig. 13k−m). Taken together, these findings suggested that the specific epithelial origin of each

type of TET may lead to a distinctive dysfunction of epithelial-cell-T-cell interactions and consequently to T-cell developmental dysfunction.

## Tumor cells induce a CD8⁺ T_RM cell-mediated immune response in type 3 TETs

Among the three types of TETs, it was interesting that precursor T cells were almost absent in type 3 TETs, whereas mature T cells were still abundant (Fig. 2c, f, g). A possible reason for this phenomenon was the immune recruitment induced by malignant cells, which is commonly observed[30]. We confirmed that CD8⁺ T_RM cells, which play a central role in the immune surveillance network of peripheral tissues[31], were significantly enriched in type 3 TETs (Fig. 6a, b). The unique abundance of CD8⁺ T_RM cells in type 3 TETs was further confirmed by flow cytometry analysis (Supplementary Fig. 14a). Compared with other epithelial subpopulations, *CHI3L1*⁺ mTEC-like cells, which were most highly enriched in type 3 TETs, were predicted to have the largest number of interactions with CD8⁺ T_RM cells (Fig. 6c and Supplementary Data 4). Specific ligand–receptor interactions between *CHI3L1*⁺ mTEC-like tumor cells and CD8⁺ T_RM cells were identified (Fig. 6d and Supplementary Data 4). The higher expression level of CXCR3 on CD8⁺ T_RM cells was confirmed by flow cytometry (Fig. 6e, f). CD8⁺ T_RM cells in type 3 TETs also expressed high levels of activation- and proliferation-related molecules (Fig. 6g and Supplementary Fig. 14b). Moreover, genes related to activation were also significantly upregulated in CD8⁺ T_RM cells compared with other types of T cells in the TETs (Fig. 6h). At the protein level, CD8⁺ T_RM cells expressed higher levels of activation-related molecules than other T-cell subsets (Fig. 6i, j and Supplementary Fig. 14c). Pathway analyses revealed that the upregulated genes of CD8⁺ T_RM cells were significantly enriched in the TCR signaling pathway, antigen processing and presentation, cytokine–cytokine receptor interaction and the chemokine signaling pathway (Fig. 6k). Further analysis demonstrated that CD8⁺ T_RM in type 3 TETs could potentially recruit T cells, B cells, DCs and macrophages via multiple chemokines (Fig. 6l and Supplementary Data 4). Through flow cytometry, we confirmed that CD8⁺ T_RM cells could secrete more of the chemokine CXCL13 than other T-cell subsets (Fig. 6m, n). Immunofluorescence staining further confirmed the colocalization of epithelial cells and CD8⁺ T_RM cells, as well as CD8⁺ T_RM cells and DCs and B cells, in type 3 TETs (Fig. 6o, p and Supplementary Fig. 14d). These results suggested that *CHI3L1*⁺ mTEC-like tumor cell-induced CD8⁺ T_RM cell activation plays a central role in the immune response and the formation of the immune landscape in type 3 TETs.

## The molecular markers of tumor cell correlate with survival of human TETs

Finally, we tried to establish a molecular classification system for human TETs based on the epithelial origin of tumor cells. First, we selected representative genes that represented the unique epithelial cell type in each TET type based on scRNA-seq results (Fig. 5f) and found that *GNB3* and *CHI3L1* were highly expressed in tumor cells of type 1 and type 3 TETs, respectively (Fig. 7a, b). Consistently, the

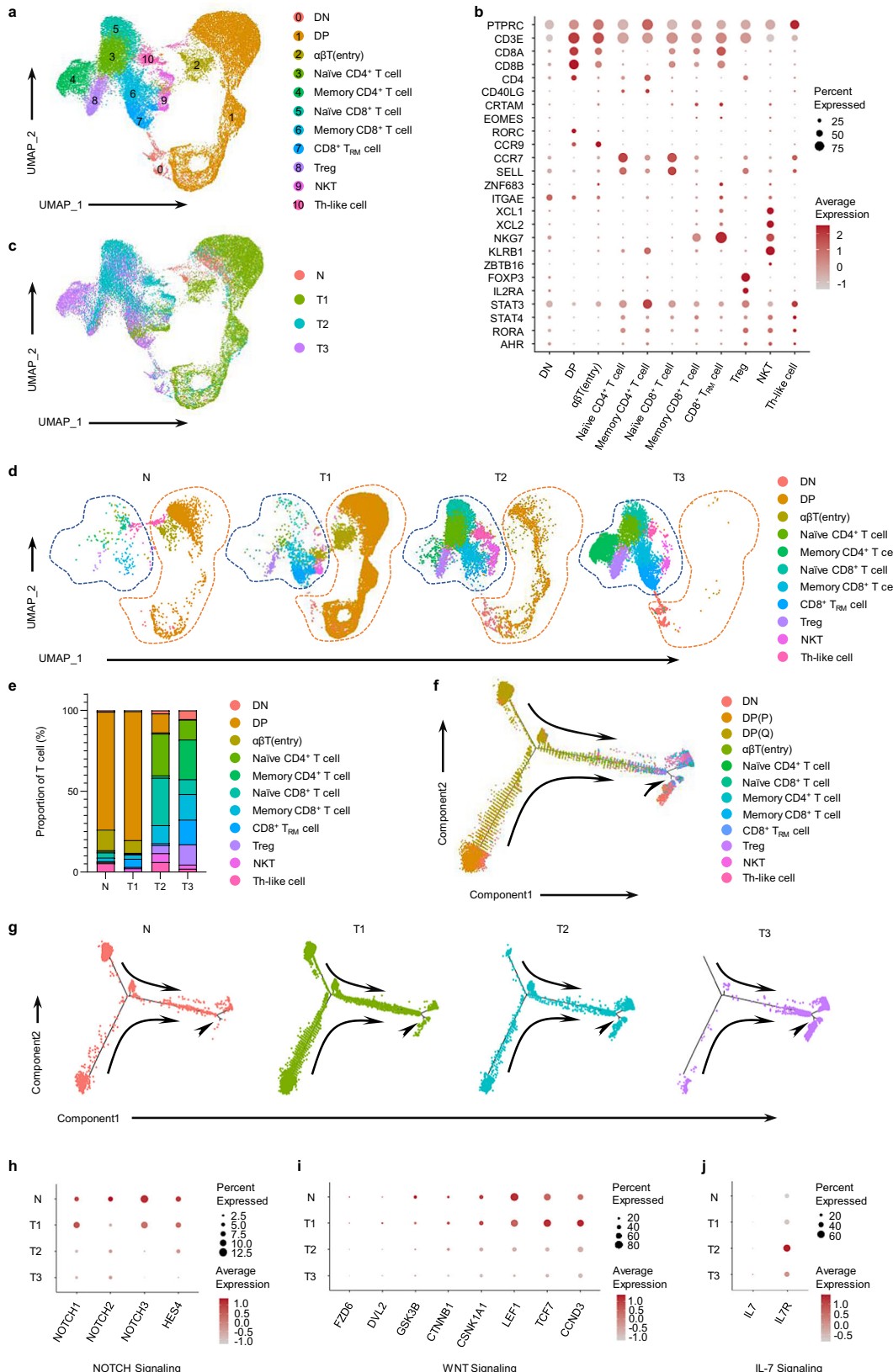

expression of *GNB3* and *CHI3L1* in tumors was further validated by IF staining (Fig. 7c). Therefore, we reclassified 119 TET patients from the TCGA cohort into three groups according to the expression levels of *GNB3* and *CHI3L1* (Supplementary Fig. 15a). Kaplan–Meier survival analysis demonstrated that the molecular classification we established was closely correlated with the survival of TET patients (*P* = 0.01409)

(Fig. 7d). However, the survival prediction effects of Masaoka staging system and WHO classification were not significant (*P* = 0.1698, *P* = 0.2912) (Supplementary Fig. 15b, c). Collectively, based on our findings, we proposed a tripartite framework to explain the origin of tumor cells and their impact on T-cell development in the tumor microenvironment of human TETs (Fig. 7e).

**Fig. 3 | T-cell development dysfunction in the thymus and TET subtypes. (a)**
Uniform manifold approximation and projection (UMAP) visualization of total
T cells from the normal human thymus (n = 1) and TET samples (n = 6), colored by
the identified cell subpopulation. (DN, double-negative T cells; DP, double-positive
T cells; the abbreviations used below are consistent with these definitions). **(b)** Dot
plot of marker gene expression in T-cell types. Here and in later figures, the color
represents the maximum normalized mean expression of marker genes in each cell
subgroup, and the size indicates the proportion of cells expressing marker genes.
**(c)** Same UMAP plot as **(a)**, colored by groups. **(d)** Same UMAP plots of total T cells
from samples of each group, colored by the identified cell subpopulation as in **(a)**.

The immature T cells are indicated by yellow dotted circles, and mature T cells are
indicated by blue dotted circles. **(e)** Composition of the T-cell compartment
showing average frequencies of major T-cell subpopulations for each group (n = 1,
2, 2 and 2 for N, T1, T2 and T3, respectively). **(f)** Pseudotime trajectory for T cells
from all samples in a two-dimensional state-space defined by Monocle2, colored by
identified T-cell subpopulation. **(g)** The same pseudotime plots for T cells from
samples of each group, colored by group. **(h-j)** Dot plots depicting the relative
expression levels of selected NOTCH **(h)**, WNT **(i)**, or IL-7 **(j)** signaling pathway
genes in total T cells from each group. Source data are provided as a Source
Data file.

## Discussion

TETs are heterogeneous tumors considered to originate from TECs[32].
However, the exact cellular origin of TETs is largely unknown[9]. Grow-
ing evidence has shown that TECs in the human thymus are extra-
ordinarily heterogeneous[25]. Therefore, it is reasonable to speculate
that the epithelial origin of human TETs may also be heterogeneous.
The findings of discrepant tumor cell compositions among human TET
types in our study supported this hypothesis. In addition, we identified
the exact tumor cell subset of each human TET type that represented
its epithelial origin. These findings suggested that there was a
remarkable heterogeneity in the epithelial origin of human TETs,
which may determine the biological nature and prognosis of
human TETs.

Genetic aberrations were reported in human TETs a decade ago[33].
Subsequently, a specific missense mutation in *GTF2I* was demon-
strated to occur at high frequency in type A and type AB thymomas but
rarely in the aggressive subtypes[34]. In contrast, thymic carcinomas
carry a higher number of recurrent mutations of known cancer genes,
such as *TP53*, *CYLD*, *CDKN2A*, *BAP1* and *PBRM1*, than thymomas[34].
Consistently, another study identified recurrent somatic mutations in
*TET2*, *CYLD*, *SETD2*, *TP53*, *FBXW7*, *HRAS* and *RB1* in thymic carcinoma
and no mutations in *GTF2I*[35]. However, the details of how these driver
oncogenes promote tumor development and the key driver genes in
each type of TET are still unknown. Recently, *GTF2I* mutation was
reported to induce cell transformation and metabolic alterations in
TECs, which provided direct evidence of the role of *GTF2I* mutation in
promoting tumorigenesis of TETs[36]. By decoding the epithelial origin
of each human TET type, we identified *GNB3* and *CHI3L1*, which were
specifically expressed in type 1 and type 3 tumors, respectively. This
evidence suggests that *GNB3* and *CHI3L1* may be candidate driver
oncogenes in the original TECs of type 1 and type 3 human TETs,
respectively.

There are various histological classifications and clinical staging
systems for human TETs, including the WHO classification and
Masaoka staging system[9]. Despite improvements in these classification
systems, some longstanding controversies remain unresolved[37–39].
Recently, genomic analysis of tumors showed that according to
mutations in *GTF2I*, T-cell signaling mRNA signatures and somatic copy
number alteration (SCNA) levels, human TETs could be classified into
four types associated with disease-free and overall survival of
patients[15]. A pioneer study uncovered the integrated genomic land-
scape of TETs and established new molecular classifications based on
the gene mutation rates and expression levels in the tumor mass,
which could predict outcomes and the incidence of autoimmune dis-
eases in TET patients[16]. Here, we proposed an alternative molecular
classification for human TETs according to the molecules specifically
expressed in the unique tumor cell subset that reflected the epithelial
origin of each tumor type. Although our proposed molecular classifi-
cation showed promise in prognostic prediction in a cohort of 119 TETs
patients, further clinical validation was needed.

A high incidence rate of autoimmune disease is one of the most
remarkable clinical features of human TETs[10]. However, the cellular
and molecular mechanisms of immune disorders in TETs are still
largely unclear. We hypothesized that the epithelial origin of human

TETs, which plays a critical role in thymic T-cell development and
central self-tolerance[13], accounted for the generation of immune
disorders in patients. As expected, we found that the abundance of
cTEC-like cells increased in type 2 TETs; in contrast, that of mTEC-
like cells decreased. Consistently, AIRE, the key molecule for the
negative selection of T cells, was not expressed in the major subsets
of tumor cells in type 2 TETs. This evidence suggested that the
increased abundance of cTEC-like cells, decreased abundance of
mTEC-like cells and decreased AIRE expression in type 2 TETs led to
the dysfunction of central self-tolerance and the generation of
autoimmune disease in patients. In addition, the finding of a relative
lack of mature T cells in tumors suggested that defects in T-cell
maturity in tumors may account for the generation of acquired
T-cell deficiency in type 1 TETs. The findings obtained in our study
provide a new clue for understanding the generation of immune
disorders in TET patients.

Taken together, we report a comprehensive study that is the first
to use multiple omics approaches to decode the biological char-
acteristics of human TETs, from the immune phenotype to the
underlying mechanism determined by the epithelial origin of tumors.
We uncovered an epithelium-T-cell loop that reflected the core dif-
ferences in biological characteristics among tumor types. Finally,
according to the immune landscape of tumors and molecular markers
of tumor cell, we proposed a potential alternative classification of
human TETs. Although the sample size of our study is substantially
smaller than the canonical classifications rely on, we believe that our
findings provide insights into the biology of human TETs and will
support translational research on human TETs.

## Methods
### Patients and specimens
Tumor tissues (T, homogeneous cellularity, without necrotic foci)
were obtained from patients with TETs who underwent surgical
resection at the Department of Thoracic Surgery, Second Affiliated
Hospital, Zhejiang University School of Medicine. None of the
patients had received radiotherapy or chemotherapy before sur-
gery. The pathological results of all patients indicated TETs. Normal
human thymus samples were obtained from the Department of
Cardiac Surgery, Second Affiliated Hospital, Zhejiang University
following cardiothoracic surgery on adults with heart disease, as
thymic tissue is routinely removed and discarded to achieve ade-
quate exposure of the retrosternal operative field. All samples were
anonymously coded in accordance with local ethical guidelines (as
stipulated by the Declaration of Helsinki), written informed consent
was obtained, and the protocol was approved by the Review Board
of the Second Affiliated Hospital of Zhejiang University School of
Medicine.

### Preparation of cell suspensions
Freshly excised tissues were stored in sterile RPMI (Corning) supple-
mented with 10% FBS (Life Technologies) and 1% streptomycin and
penicillin (Life Technologies) and processed within 2 hours. The tis-
sues were cut into small pieces and then digested in RPMI containing
10% FBS, type I collagenase (1 mg/ml), and type IV collagenase

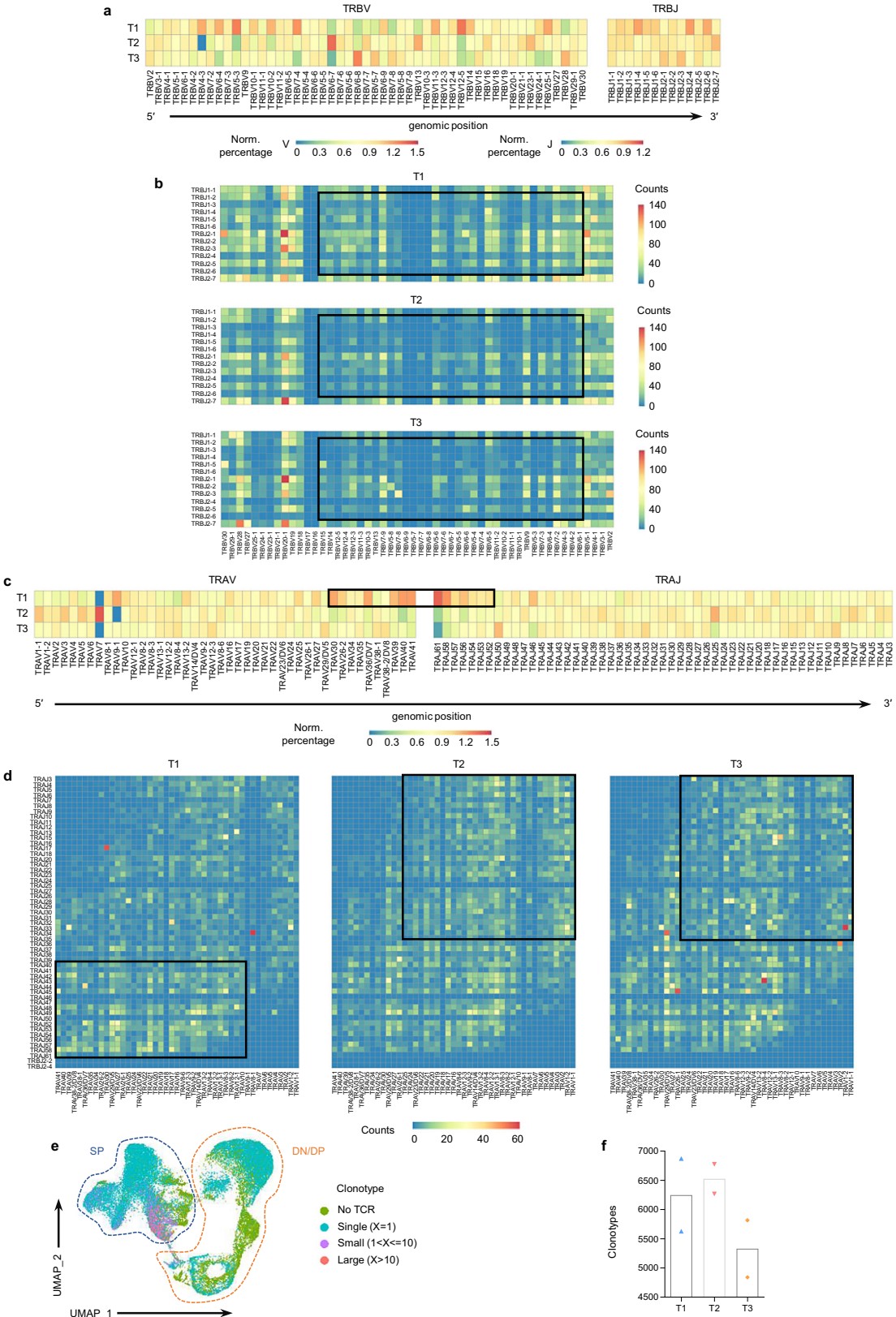

**Fig. 4 | VJ gene usage and pairing and TCR diversity in TET subtypes. (a)**
Heatmap showing the proportion of each TCRβ V and J gene segment present in total T cells from each group, normalized per column. Gene segments positioned according to genomic location. (**b**) Frequency of V-J gene pairs at the TCRβ locus of each group. (**c**) Same scheme as in (**a**) applied to TCRα V and J gene segments,
normalized per column. (**d**) Same scheme as in (**b**) applied to TCRα V-J gene pairs. (**e**) UMAP plot of all T cells from TET samples (n = 6), colored by the number of TCR clonotypes detected. (**f**) Bar plot showing clonotypic diversity among the three groups of TET samples (n = 2, 2 and 2 for T1, T2 and T3, respectively). Source data are provided as a Source Data file.

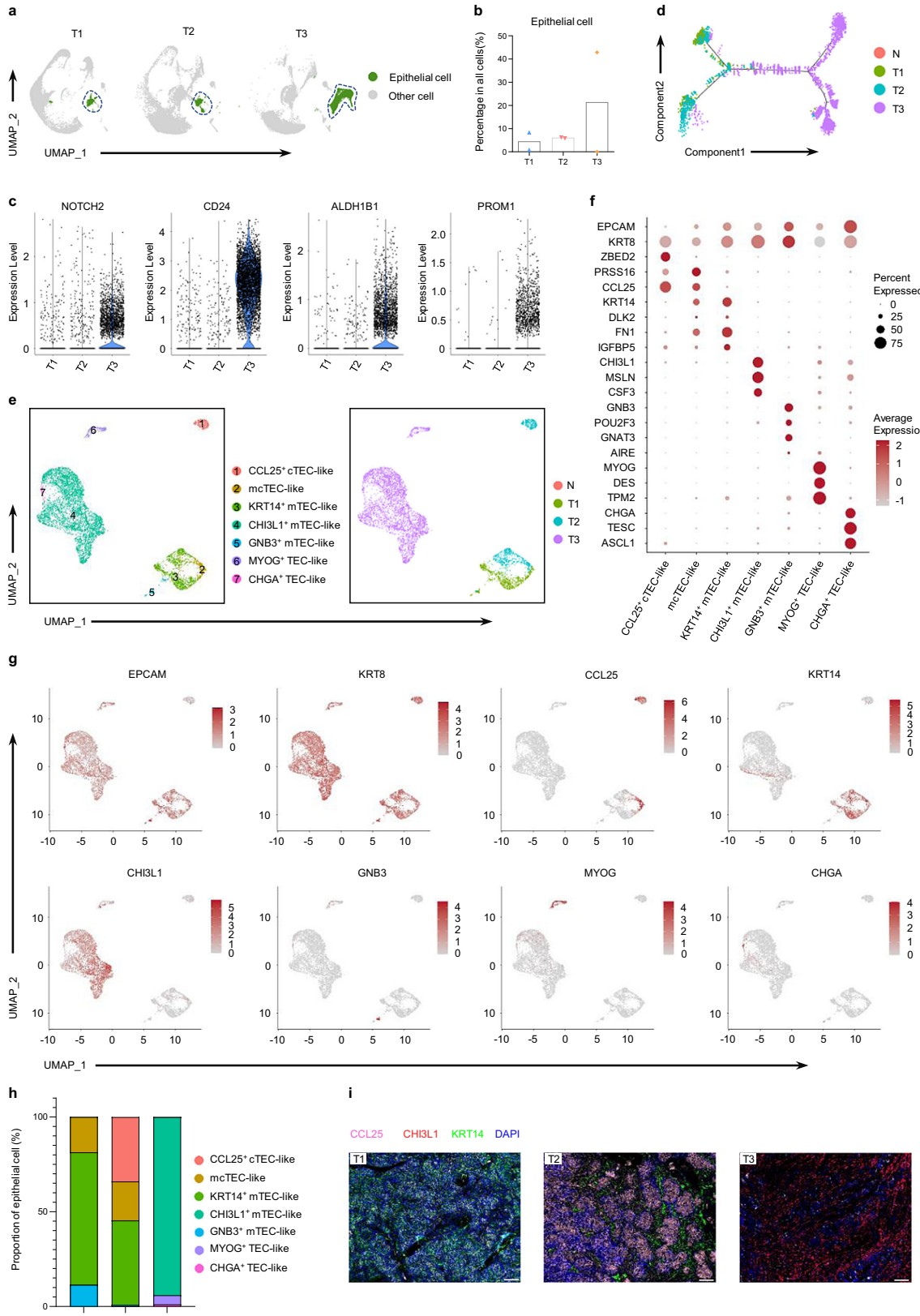

(1 mg/ml) for 1 hour at 37 °C using a gentleMACS™ Dissociator (Miltenyi Biotech) according to the manufacturer's instructions. The resulting single-cell suspension was filtered sequentially through sterile 70 μm cell strainers. Then, the cell suspensions were stored in complete medium at 4 °C for subsequent experiments.

### HE staining and image acquisition

For HE analyses, TETs and normal thymus tissues were fixed in 10% formalin, embedded in paraffin and sectioned transversely. The 5 μm-thick formalin-fixed paraffin-embedded (FFPE) sections on glass slides were incubated at 37 °C overnight, dewaxed in xylene, rehydrated

**Fig. 5 | Heterogeneity of epithelial cells among the three TET tumor types. (a)** UMAP visualization of total cells from samples in each group, colored by the cell types of epithelial cells and other cells (n = 2, 2 and 2 for T1, T2 and T3, respectively). **(b)** Bar plot showing the frequencies of epithelial cells among the TET groups (n = 2, 2 and 2 for T1, T2 and T3, respectively). **(c)** Violin plots showing the expression of the marker genes for CSCs on epithelial cells among different types of TETs. **(d)** Pseudotime trajectory for epithelial cells from all samples in a two-dimensional state-space defined by Monocle2, colored by group. **(e)** UMAP plot of all epithelial cells from the normal human thymus (n = 1) and TET samples (n = 6), colored by the identified cell subpopulation (left). Same UMAP plot, colored by group (right). **(f)** Dot plot of the expression of marker genes in epithelial cell types. **(g)** UMAP visualization of the expression of marker genes used for epithelial cell cluster identification. The color scale represents normalized expression. Gray to brown: low to high expression. **(h)** Composition of the epithelial cell compartment showing average frequencies of major epithelial cell subpopulations for each group of TETs; each color represents an epithelial cell subset (n = 2, 2 and 2 for T1, T2 and T3, respectively). **(i)** Representative IF staining images showing CCL25 (pink), CHI3L1 (red), KRT14 (green) and DAPI (nuclei, blue) in TET samples (scale bar: 100 μm). Experiment was performed in three independent samples for each group with similar results. Source data are provided as a Source Data file.

through decreasing concentrations of ethanol, and washed in PBS. Then, the sections were stained with hematoxylin and eosin. After staining, the sections were dehydrated through increasing concentrations of ethanol and xylene. The slides were scanned with a KF-PRO-120 scanner (KFBIO).

### Immunofluorescence staining and image acquisition
The 5 μm-thick FFPE sections were also incubated at 37 °C overnight, dewaxed in xylene, rehydrated through decreasing concentrations of ethanol, and washed in PBS. Antigen retrieval was performed in Target Retrieval Solution (TRS) citrate buffer (pH 6.0) using a pressure cooker. Then, the sections were blocked with a blocking buffer solution (5% FBS, 1% BSA and 0.2% Triton) for 2 h at room temperature and incubated with the primary antibody in blocking buffer (BioLegend) at 4 °C overnight. After washing with PBS (pH 7.4), polymer horseradish peroxidase (HRP) conjugated secondary antibody staining was performed at room temperature for 1 h. Tyramide signal amplification (TSA)-based visualization was performed with Opal fluorophores. Next, the sections were washed three times with PBST, and DAPI reagent was added for 10 min to detect cell nuclei. Antibodies against CD3 (ab16669, diluted 1:100), CD8a (ab17147, diluted 1:100), CD4 (ab133616, diluted 1:500), CD103 (ab129202, diluted 1:800), CD11c (ab52632, diluted 1:500), CD20 (ab78237, diluted 1:2000) and EPCAM (ab223582, diluted 1:500) were procured from Abcam. Antibodies against CD45 (60287-1-IG, diluted 1:500), CCL25 (25285-1-AP, diluted 1:500), CHI3L1 (12036-1-AP, diluted 1:400), AIRE (22517-1-AP, diluted 1:500), KRT14 (60320-1-IG, diluted 1:800) and GNB3 (12036-1-AP, diluted 1:400) were procured from Proteintech Group. HRP conjugated Goat Anti-Mouse/Anti-Rabbit secondary antibodies (ab2891, 1:200) were procured from Abcam. Epifluorescence multispectral whole-slide images of all sections were acquired through the NIKON ECLIPSE C1 system (Nikon Corporation) and scanned at high resolution on a Pannoramic SCAN II system (3DHISTECH Ltd.).

### Flow cytometry
The antibodies CD45-BV510 (2D1, diluted 1:100), CD45-APCCY7 (HI30, diluted 1:100), CD3-BV421 (UCHT1, diluted 1:100), CD3-BV605 (UCHT1, diluted 1:100), CD3-APCCY7 (UCHT1, diluted 1:100), CD3-PECY7 (UCHT1, diluted 1:100), CD4-APC (RPA-T4, diluted 1:100), CD4-PECY7 (RPA-T4, diluted 1:100), CD8a-APCCY7 (RPA-T8, diluted 1:100), CD8a-AF700 (RPA-T8, diluted 1:100), CD8a-FITC (RPA-T8, diluted 1:100), CD103-BV605 (Ber-ACT8, diluted 1:100), CD103-FITC (Ber-ACT8, diluted 1:100), CD69-PECY7 (FN50, diluted 1:100), CD69-PerCP-Cy5.5 (FN50, diluted 1:100), CD39-PE (A1, diluted 1:100), IFN-γ-PE (B27, diluted 1:50), CXCR3-BV421 (G025H7, diluted 1:100), EPCAM-PE (9C4, diluted 1:100), EPCAM-APC (CO17-1A, diluted 1:100), and 7-AAD (Cat#:420404, diluted 1:200) were procured from BioLegend. CXCL13-APC (Cat#: IC801A, diluted 1:20) was procured from R&D Systems. We preincubated fresh tissue cells (1×10⁶/ml) in a mixture of PBS, 2% fetal calf serum, and 0.1% (w/v) sodium azide with FcgIII/IIR-specific antibody to block nonspecific binding and stained them with different combinations of fluorochrome-coupled antibodies for 15 minutes at room temperature. IFN-γ and CXCL13 were stained for 30 minutes at room temperature after fixation and permeabilization.

Cells were washed with PBS and passed through a 70-μm filter, and data were collected on a FACSCanto II system and FACSFortessa system (BD Biosciences) and analyzed using FlowJo software (version 10.0.7). Gating strategies used for the flow cytometry were presented in supplementary Fig. 16.

### Mass cytometry (CyTOF) sample preparation
The 42 metal-conjugated antibodies used in this study are shown in Supplementary Table 1. Briefly, the cells derived from the TET samples were stained with 5 μM cisplatin (Fluidigm) in PBS without BSA for viability staining. Then, the samples were washed in PBS containing 2.5% BSA and blocked for 30 minutes at 4 °C. After that, they were stained with cell-surface antibodies in PBS containing 5% goat serum and 30% bovine serum albumin (BSA) for 30 min at 4 °C. Next, the samples were washed, fixed and permeabilized using the Foxp3 fix and permeabilization kit (eBioscience) as well as 100 nM Iridium nucleic acid intercalator (Fluidigm) according to the manufacturer's instructions at 4 °C overnight. The cells were then washed twice with Foxp3 permeabilization buffer and incubated with intracellular antibodies in permeabilization buffer for 30 min at 4 °C. Finally, the cells were washed twice with ddH₂O to prepare them for analysis.

### CyTOF data acquisition and analysis
Immediately before the acquisition, samples were washed and resuspended at a concentration of 1 million cells/ml in water containing EQ Four Element Calibration Beads (Fluidigm). Samples were acquired on a Helios CyTOF System (Fluidigm) at an event rate of <500 events/second. EQ beads (Fluidigm) were used as a loading control. All data were produced on a Helio3 CyTOF Mass Cytometer (Fluidigm). Mass cytometry data files were normalized using bead-based normalization software, which uses the intensity values of a sliding window of bead standards to correct for instrument fluctuations over time and between samples. CyTOF analyses were performed by PLTTech Inc. (Hangzhou, China) according to a previously described protocol[40]. The data were gated to exclude residual normalization beads, debris, dead cells and doublets for subsequent clustering and high dimensional analyses. Forty-two immune cell markers were all applied for clustering and visualization. The Phenograph[41], PARC and Xshif algorithms were used to cluster the cells. A total of 50,000 cells were selected randomly for visualization by the t-distributed stochastic neighbor embedding (t-SNE) dimension reduction algorithm[42] using the R package cytofkit (version 0.13). Immune subset cells were defined by the median values of specific expression markers in hierarchical clustering. Heatmaps of normalized marker expression, relative marker expression, and relative differences in population frequency were generated using the pHeatmap R package and Python (https://www.python.org/). Comparisons between two groups were performed by unpaired Student's t tests using GraphPad Prism (v8). The use of these tests was justified based on an assessment of the normality and variance of the distribution of the data.

### Single-cell RNA sequencing
Trypan blue was used for quality evaluations of single-cell suspensions prepared as outlined earlier, and the cell survival rate was

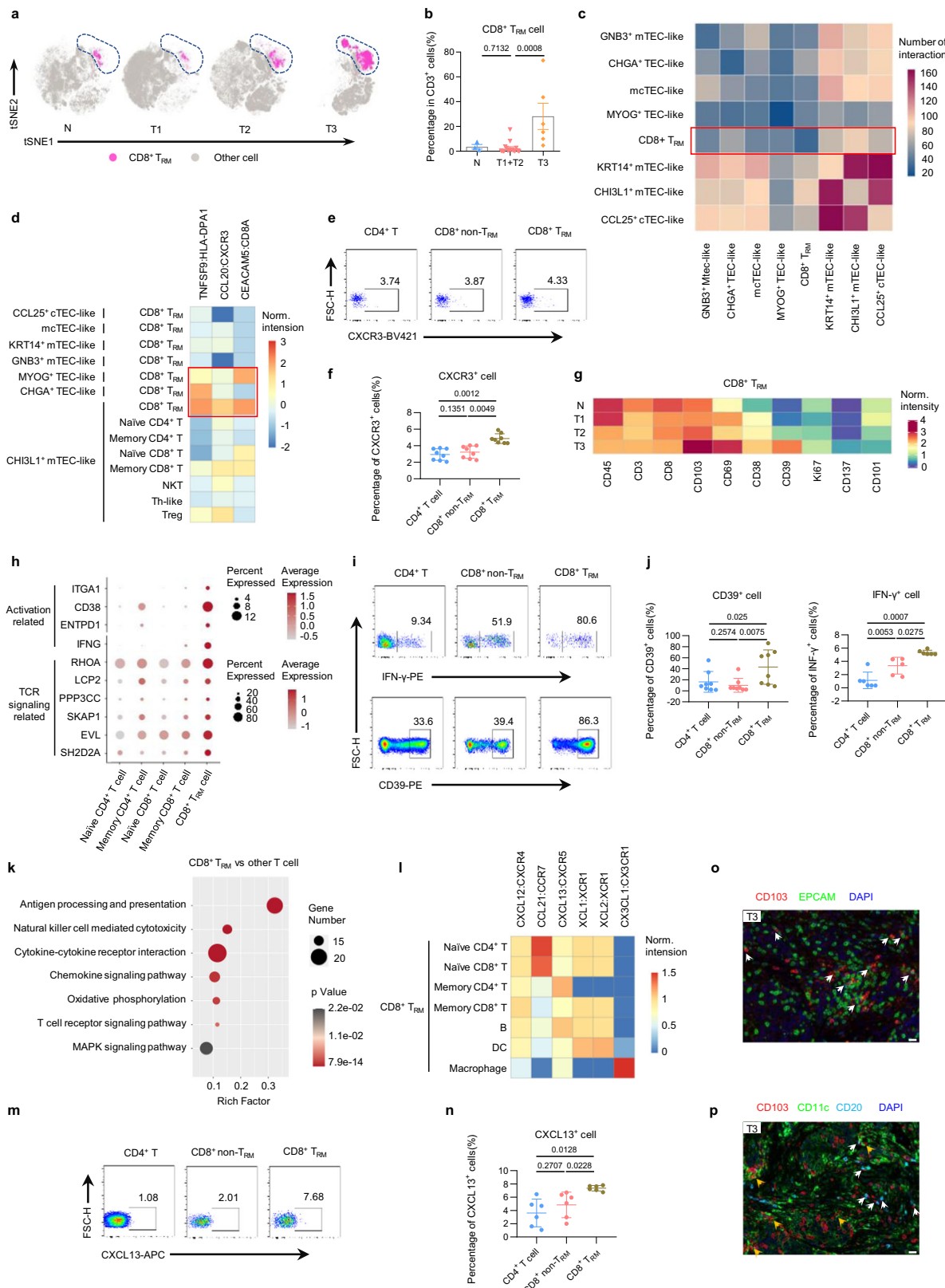

generally above 80%. The cells that passed the test were washed and resuspended to prepare a suitable cell concentration of 1000 cells/μl for 10x Genomics Chromium™. Approximately 10,000 cells were loaded onto the 10X Chromium Single Cell Platform (Single Cell 5′ library and Gel Bead Kit v.3) as described in the manufacturer's protocol. The generation of gel beads in emulsion (GEMs),

barcoding, GEM-RT clean-up, complementary DNA amplification and library construction were all performed according to the manufacturer's protocol. Qubit (Agilent 2100) was used for library quantification before pooling. The final library pool was sequenced on an Illumina NovaSeq 6000 instrument using 150-base-pair paired-end reads.

**Fig. 6 | The mechanism of tumor-driven CD8+ T_RM cell activation in type 3 TETs.** (**a**) Same t-SNE plots as (Fig. 2c), colored by CD8+ T_RM cells and other cells. (**b**) Bar plot showing the frequencies of CD8+ T_RM cells among the three groups (n = 3, 16 and 6 for N, T1 + T2 and T3, respectively. Data are presented as the mean ± s.e.m. *P* values in the figure were determined by an unpaired two-tailed Student's t test). (**c**) Heatmap showing the number of interactions between CD8+ T_RM cells and epithelial cell subpopulations. (**d**) Heatmap of chemokine interactions among T cells and epithelial cells, normalized per column. (**e–f**) Flow cytometric analysis of CXCR3 expression among CD4+ T cells, CD8+ non-T_RM cells and CD8+T_RM cells in TETs. (**g**) Heatmap showing the expression of activation-associated proteins in CD8+ T_RM cells in each group by CyTOF and normalized total matrix. (**h**) Dot plot of the expression of cell activation- and TCR activation-associated genes within CD8+ T_RM cells and other T-cell subsets. (**i-j**) Flow cytometric analysis of IFN-γ and CD39 expression among CD4+ T cells, CD8+ non-T_RM cells and CD8+T_RM cells in TETs. (**k**)

KEGG pathway analysis of genes that were significantly (*P* < 0.01) upregulated in CD8+ T_RM cells compared to other T cells. (**l**) Heatmap of chemokine interactions among CD8+ T_RM cells and other cell types, normalized per column. (**m–n**) Flow cytometric analysis of CXCL13 expression among CD4+ T cells, CD8+ non-T_RM cells and CD8+ T_RM cells in TETs. (**o–p**) Representative IF staining images showing CD103 (red), EPCAM (green), CD11c (green), CD20 (pale blue) and DAPI (nuclei, blue) in type 3 TET samples. The interactions between epithelial cells and T_RM cells are indicated by white arrows (**o**). The interactions between T_RM cells and DCs are indicated by yellow arrows, and those between T_RM cells and B cells are indicated by white arrows (**p**). Scale bar: 20 μm. Experiment was performed in three independent samples with similar results. (**f, j, n**) n = 8(**f**), 8 (**j-left**), 6 (**j-right**), 6(**n**), respectively. Data are presented as the mean ± s.e.m. *P* values in the figure were determined by a two-tailed paired Student's t test. Source data are provided as a Source Data file.

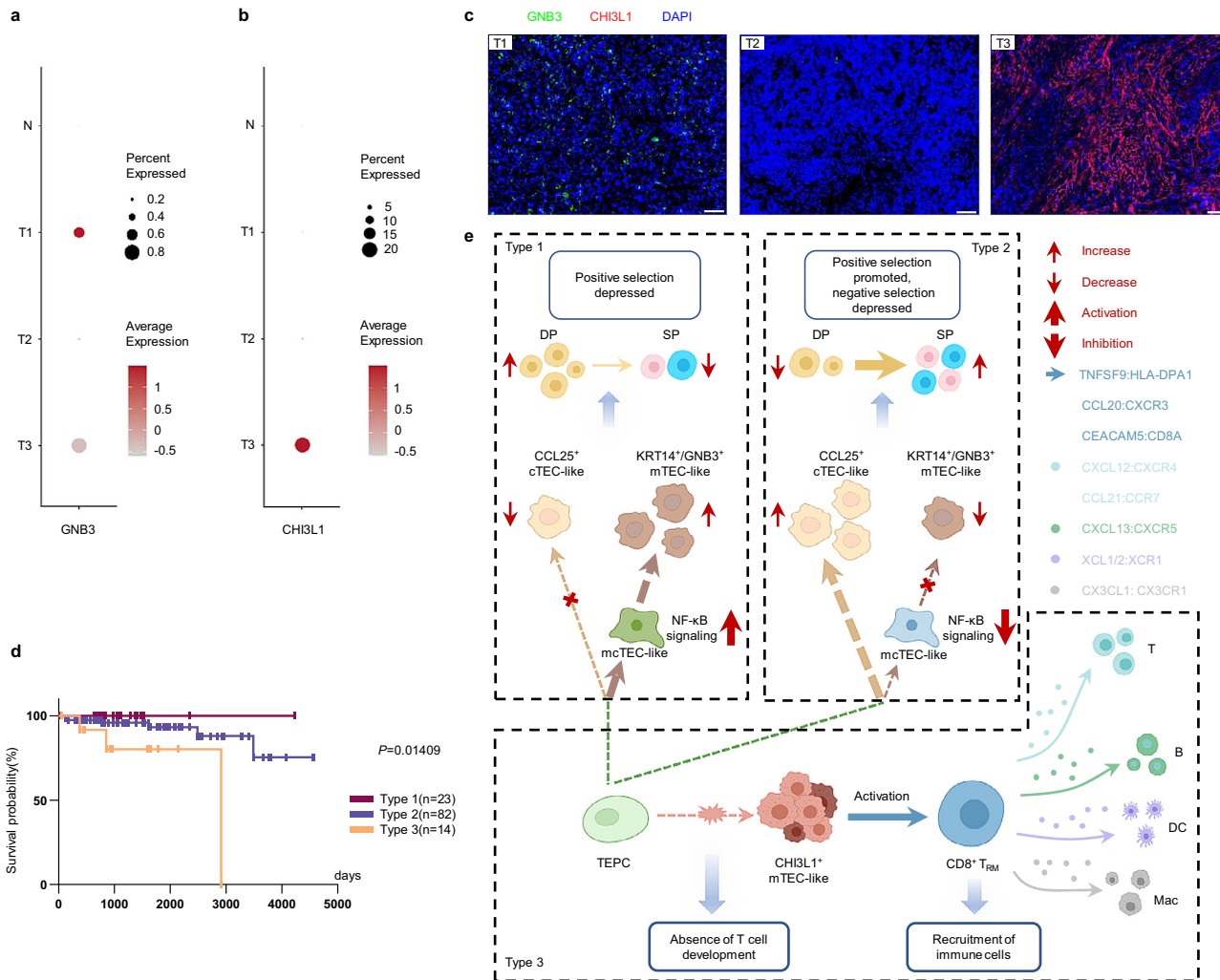

**Fig. 7 | Alternative classification of TETs using representative gene markers of tumor cells.** (**a-b**) Dot plots depicting the relative levels of the *GNB3* (**a**) and *CHI3L1* (**b**) genes in total cells from each group. (**c**) Representative IF staining images showing GNB3 (green), CHI3L1 (red), and DAPI (nuclei, blue) in TET samples. Scale bar: 50 μm. Experiment was performed in three independent samples for each group with similar results. (**d**) Overall survival (OS) examined in each of the three types of TETs in the TCGA cohort. The *P* value was calculated using the log-rank test. (**e**) Schematic model summarizing the developmental patterns of three types of TETs, including interactions between different types of cells. The ligand is secreted by the cell at the beginning of an arrow, and the receptor is expressed by the cell at the end of that arrow. Source data are provided as a Source Data file.

## Single-cell gene expression analysis

Raw gene expression matrices were generated using CellRanger (10x Genomics) and analyzed using the Seurat v3 R package[43,44]. All cells expressing <500 genes were removed, as well as cells that contained <500 unique molecular identifiers (UMIs) and >25% mitochondrial counts. Samples were merged and normalized. Because every cell has a unique barcode, the scRNA-seq data could be linked with the scTCR-seq data.

## scRNA-seq clustering analysis

The LogNormalize method of the "Normalization" function of Seurat software was used to calculate the expression values of genes.

Principal component analysis (PCA) was performed using the normalized expression value. Among all the PCs, the top 10 PCs were used to perform clustering and t-SNE analysis. We used the shared nearest neighbor (SNN) graph-based clustering algorithm from the Seurat package (FindClusters function with default parameters) to identify clusters of cells. Among them, 18 clusters demonstrated high expression of genes specific to epithelial and T cells and were thus selected for the final reclustering analysis. These clusters were then identified as distinct cell subtypes based on highly expressed genes from a literature review.

### Differential expression and pathway analysis
DEGs were identified using the bimod test with the FindMarkers and FindAllMarkers functions in Seurat with a threshold of $\log_2$ (fold change) ($\log_2$FC) $\geq 0.26$ and expression in more than 10% of cells. In addition, we selected 0.01 as the cutoff of the p value. KEGG[45] analyses were performed using the OmicStudio tools (https://www.omicstudio.cn/tool) to identify gene set enrichment with a hypergeometric test.

### TCR analysis
Full-length TCR V(D)J segments were enriched from amplified cDNA via PCR amplification using a Chromium Single-Cell V(D)J Enrichment kit according to the manufacturer's protocol (10X Genomics). 10X TCR-enriched libraries were mapped with the Cell Ranger Single-Cell Software Suite (version 4.0.0, 10x Genomics) to the custom reference provided by the manufacturer (version 2.0.0 GRCh38 VDJ reference). All assembled contigs were filtered to retain only those that were assigned a raw clonotype ID and categorized as being both full-length and productive. Each clonotype was assigned a unique identifier, consisting of the predicted amino acid sequences of the CDR3 regions of these two chains, which was used to match clonotypes across samples. Clonality, which reflects the dominance of particular clones across the TCR repertoire, was calculated for each sample. To visualize the proportion of TCR clonotypes shared between T-cell phenotypes, we used barcode information to project T cells with prevalent TCR clonotypes on UMAP plots.

### Pseudotime reconstruction and trajectory inference
The R package Monocle (version 2) algorithm was used to reconstruct pseudotime trajectories to determine the potential lineage development among diverse T-cell subsets and epithelial cell subsets[46]. For each analysis, PCA-based dimension reduction was performed on DEGs of each phenotype, followed by two-dimensional visualization on UMAP. Graph-based clustering (Louvain) sorted the T cells into twelve subsets. The cell differentiation trajectory was then captured using the orderCells function. As described previously for differential gene expression analyses, the R package MAST was used to detect genes significantly covarying with pseudotime based on a log-likelihood ratio test between the model formula, including cell pseudotime, and a reduced model formula. Additional model covariates were included in the residual model formula. Benjamini–Hochberg multiple testing correction was used to calculate the FDR, and genes with a FDR < 5% were considered to vary significantly with pseudotime. For T cells from different groups, the same process with the same signature genes and Monocle parameters was used to construct the clone-based trajectories.

### RNA velocity analysis
The spliced and unspliced reads were counted from aligned bam files generated by CellRanger and fed into the velocyto.R (version 0.6) to calculate RNA velocity values for each gene in each cell. The resulting RNA velocity vector was then embedded into the UMAP space.

### Cell-to-cell communication analysis of scRNA-seq data
CellPhoneDB (www.CellPhoneDB.org) was used to assess putative receptor–ligand interactions between epithelial cells and T-cell subsets, CD8$^+$ T$_{RM}$ cells and other cell subsets (epithelial cells, B cells, DCs, fibroblast cells, monocytes, macrophages, and VSMCs). Briefly, the algorithm allows the detection of ligand–receptor interactions between cell types in scRNA-seq data using the statistical framework described in previous studies[47,48]. The tool was run for 1000 iterations. The normalized interaction score was calculated by multiplying the average expression level of ligands and receptors for all cell pairs and maximum normalizing these values. The total number of pairwise paracrine interactions between CD8$^+$ T$_{RM}$ cell and epithelial cell subsets obtained using the CellphoneDB scoring method were visualized as heatmaps in the R package pheatmap.

### Cell isolation and in vitro coculture
After antibody staining, CD45$^-$EPCAM$^+$ epithelial cells and CD45$^+$CD3$^+$CD4$^+$CD8$^+$ T cells were sorted by an Aria II cell sorter (BD Biosciences), and 7-AAD was used to eliminate dead cells. The purity of all sorted cells was greater than 90%. CD45$^-$EPCAM$^+$ epithelial cells and CD45$^+$CD3$^+$CD4$^+$CD8$^+$ T cells were mixed in 96-well plates for coculture in the presence of rhIL-7 (10 ng/ml) (Cat#: GMP200-07). The sorted cells were cocultured at a ratio of epithelial:CD4$^+$CD8$^+$ T cells of 1:2 (25000:50000). Another group containing CD4$^+$CD8$^+$ T cells was prepared as a negative control by adding rhIL-7 alone without epithelial cells. After coculture in vitro for 8 days, the proportions of DP (CD3$^+$CD4$^+$CD8$^+$ T) cells and SP (CD3$^+$CD4$^+$ T and CD3$^+$ CD8$^+$ T) cells among CD3$^+$ T cells were analyzed by flow cytometry.

### RNA isolation and reverse-transcription quantitative PCR
Total RNA was extracted from CD3$^-$CD4$^+$CD8$^+$ and CD3$^+$CD4$^+$CD8$^+$ cells sorted by an Aria II cell sorter (BD Biosciences) using TRIzol reagent (Thermo Fisher Scientific K.K.). The cDNA was reverse-transcribed from 500 ng of total RNA using PrimeScript$^{TM}$ RT reagent Kit (Takara) according to the manufacturer's instructions. Quantitative PCR was performed on a Bio–Rad MyiQ Real-Time PCR Detection System using TB Green Premix Ex TaqTM II(Takara). The relative mRNA expression of CD3E and CD3G was normalized to the mRNA levels of the housekeeping gene GAPDH. The real-time PCR primers were synthesized by Integrated DNA Technologies. Primer sequences can be found in Supplemental Table 8.

### Survival analysis
TCGA TET data were used to evaluate the prognostic effect of individual genes or gene sets derived from specific cell clusters. The gene expression and survival data were downloaded from UCSC Xena (http://xena.ucsc.edu/). A total of 119 patients with records containing GNB3 and CHI3L1 gene expression information were included in the survival analysis (of these, 117 had a WHO classification and Mosaok stage). The expression profile was normalized by log2 (normalized_count+1) to exclude potential bias. For each gene, patient cohorts were grouped into high- and low-expression groups by the top 20% of the normalized average expression. Kaplan–Meier survival curves were plotted to show differences in survival time, and P values were reported from the log-rank test using GraphPad Prism (v8).

### Statistical analysis
Statistical analysis was performed using Prism version 8.3 (GraphPad Software). Two-sided Student's t tests were used to identify differences between two groups. All results are presented as the mean ± SEM, and P values of <0.05 were considered statistically significant.

### Reporting summary
Further information on research design is available in the Nature Research Reporting Summary linked to this article.

## Data availability

The raw scRNA-seq data reported in this paper has been deposited in the Genome Sequence Archive in National Genomics Data Center under the accession number HRA002334. For analysis of the normal thymus, Jong-Eun Park et al.'s dataset[17] were downloaded from ArrayExpress (accession number E-MTAB-8581). The remaining data were available within the Article, Supplementary Information, or Source Data file. Source data are provided with this paper.

## Code availability

Custom codes were not created for data analyses in this study. Analysis of CyTOF followed publicly available instructions from PhenoGraph (https://github.com/jacoblevine/PhenoGraph), Xshift (https://github.com/ginberg/xshift_operator) and PARC (https://github.com/ShobiStassen/PARC). Analysis of scRNA-seq followed publicly available instructions from Cellranger (https://github.com/10XGenomics/cellranger), Seurat (http://satijalab.org/seurat/), Monocle (http://coletrapnell-lab.github.io/monocle-release/docs/), Velocyto (http://velocyto.org) and CellPhoneDB (https://www.cellphonedb.org). Any additional information required for the analysis of data in this manuscript is available from the authors upon request.

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

## Acknowledgements

We are grateful to the tissue donors and the clinical staff at The Second Affiliated Hospital of Zhejiang University School of Medicine who made tissue collection possible. We thank the Jian Huang lab for technical assistance and helpful support. We thank all the members of the Pin Wu lab for discussions and comments on the manuscript. This work was supported by funding from the National Natural Science Foundation of China (81572800 and 82073141 to PW, 82073142 to DW), the Fundamental Research Funds for the Central Universities (2019QNA7025 to PW) and the Natural Science Foundation of Zhejiang Province (LY15H160041 and LR22H160006 to PW, LY19H160050 to DW).

## Author contributions

P.W. designed the study. Z.W.X., M.J.L., Z.X.H., D.C., Y.Y.C. and performed the experiments. Z.W.X., M.J.L. and Z.X.H. collected the samples and clinical data. X.X. and J.F.L. reviewed the histological types of the samples. Z.W.X., D.W. and P.W. analyzed the data, interpreted the data, and wrote the manuscript. Y.C. and P.W. supervised the project, and all authors reviewed and edited the manuscript.

## Competing interests

The authors declare no competing interests.
