## [Peer Review File · Nature Communications]

The immune landscape of human thymic epithelial tumorsREVIEWER COMMENTS

Reviewer #1 (Remarks to the Author):

In this manuscript, Xin et al. conducted a single-cell study of thymic tumors based on CyTOF and single-cell RNA sequencing. A new classification of thymic tumors was proposed and multiple characteristics were discussed. While this study is interesting, critical concerns exist before publication.

1. It is unclear how the new classification was derived. Since single-cell data including CyTOF and scRNA-seq were high-dimensional, it is expected that multiple clustering schemes exist. The validity, robustness, and significance of the current classification should be demonstrated. However, the current manuscript is mainly descriptive. Comparison with alternative clustering schemes including WHO classifications was absent, limiting the significance of this study.
2. Since CyTOF data were on the protein level while scRNA-seq data were on the RNA level, a comparison between CyTOF findings with the observations based on scRNA-seq should be explicitly presented regarding both cell proportion and molecular characteristics.
3. The discussion on the impact of tumors on T cell development is interesting. However, the current analysis seems to be superficial. As extrinsic T cells could be recruited to the thymus because of the immunogenicity of thymic tumors, the observations of defects in T cell development may result from the variance in sample collection rather than the biological impacts of tumor on T cell development. The authors should add new analyses to evaluate such possibilities.
4. The evidence supporting the epithelial origins is weak. A deeper analysis including lineage tracing based on mitochondrial mutations or RNA velocity should be added. Otherwise, independent validation experiments should be added.
5. Related to 4, it is difficult to directly derive the conclusion that epithelial origins had important impacts on the deficiency of T cell development. Alternative factors including acquired characteristics of different tumors could also contribute to the distortion of T cell development. Evidence supporting the roles of epithelial origins should be added.

Minor concerns:

6. The language should be proofread by a native English speaker.
7. The figures can be further improved for clarity.

Reviewer #2 (Remarks to the Author):

An improved classification of thymic epithelial tumors based on their molecular and cellular characteristics is important towards better understanding of those tumors and their consequences in the immune system and towards better treatment of the disease. However, this study described only 18 cases (2 A, 5 AB, 1 B1, 4 B2, 2 B3, and 3 C) in total and showed single cell RNA sequencing results only from 6 patients. Such a small number of the subjects critically diminishes the strength of this study. The new classification of three types sounds provocative but describes no objective basis other than graphical impression of CyTOF data shown in Figure 1B. Indeed, the manuscript offers no evaluation of pretexting WHO classification or Masaoka stages (what is improved in their new classification? what was wrong with the conventional technology). In addition, the manuscript offers no detailed description regarding how tumor cells and infiltrating hematopoietic cells were separated from neighboring cells, hampering to distinguish between the cells within the tumor and the cells proximal but outside the tumor. Therefore, the data shown in this manuscript are far from providing convincing or novel conclusion in terms of the re-classification and characterization of thymic epithelial tumors and associated thymocytes.

Reviewer #3 (Remarks to the Author):

The manuscript by Xin et al is generally interesting, with lots of data and potentially important clinical implications. However its major weakness is that it is purely descriptive and somewhat speculative.

The authors largely rely on expression data to make claims regarding cell-cell interactions, cell recruitment or cells' effect without providing any further proof/validations for these claims.

The other major weak point of the manuscript is the clarity of the text and of data presentation.

The text requires serious language edits with a professional English editor. Some parts of the text are hard to understand. Moreover, the figure legends lack a lot of critical information. For example, the authors do not mention if a heatmap is normalized per row/column/total matrix. This is especially problematic in 2B where it seems that it's definitely not normalized per gene, if it's normalized per cluster only (within each cluster) it's completely meaningless (different expression level of different genes within the same cells doesn't mean anything, some genes need to be expressed highly to have an effect, while others only need a few molecules to work..). I would hope that it's normalized for the entire matrix, which is a strange choice but at least it would make sense. I would suggest they add this information, and normalize per gene if they haven't done it. Another example is that they don't say what are the squares in 2B,D.

While the classification of type1 and 3 are convincing (Fig 1B), the classification of type 2 seems less convincing ... 04 seems more like normal, and 05 looks completely different from 10 and 11, which themselves look more like type1. Thus, the overall classification of the tumors as proposed by the authors is questionable. Most common tumor seems to be type 1, then type 3, there seem to be some other less common tumors like 04 and 05, which in my opinion do not fit into the 3 model classification.

Generally, it is unclear to me from the methods how the authors decided to group the samples into the 3 types.

Moreover, since the tumors are epithelial tumors, it would make more sense to present the classification based on epithelial cells from the very beginning.

Correspondingly a much better characterization of the epithelial subsets should be provided and compared with TEC subsets in healthy thymus. For instance GNB3 is a marker of tuft cells – Does this tumor express additional tuft cell markers? The characterization of Krt14+ cells is very non-informative – more markers are needed to better characterize this subset.

More phenotypic data should be presented for the patients with the individual tumor types. The data suggest that patients who developed type 3 tumor, had poorer survival. Is information about development of autoimmune diseases (e.g. myasthenia gravis, which occurs in large fraction of thymoma patients) available? Could this be correlated with the expression of acetyl choline receptor (CHRNA) in the individual types

What about other symptoms, such as immunodeficiency or immune dysregulation? Would be interesting to correlate it the type of the tumor with the specific phenotype.

-Fig. 2B- how can the DP T cells be CD45RO+?

-"To solve this divergence, we further annotated DP cells into proliferating (P) and quiescent (Q) subsets according to the expression level of MKI67 and CDK1 as previous study 21 (Supplementary Fig. S4C and S4D). Consistently, the marker genes of CD3- DP cells, which were obtained from the protein marker-based sorting strategy in another study 23, were obviously highly expressed in DP (P) cells (Supplementary Fig. S4E). These results suggested a close match between the DP (P) cells defined by scRNA-seq and CD3- DP cells detected by CyTOF in our study." I didn't really understand authors' explanation about the discrepancy between the RNA expression and CyTOF data for CD3 in DP cells- how does proliferation explain why CD3-expressing DP cells don't bind the antibody for CD3?

6I – The authors show the cells are in proximity to epithelial cells but this is very general, are they close to the CHI3L1 cells in particular?

-in the 6th chapter, could they provide an actual proof, any additional validation other than expression data for their claims? For example the activation of CD8 Trm cells or their recruitment of other immune cells? Ex vivo even... Or even just the actual protein overexpression of the ligands raised in 6G... Or for the ligand-receptor interactions between the CD8 Trm cells to CHI3L1 epithelial cells...I think any additional validation will strengthen their paper, this chapter feels a bit speculative as it is.

April 30, 2022

Editorial Office
Nature Communications

The enclosed manuscript entitled “Molecular imprint of epithelial origin identifies immuno-subtype of thymic epithelial tumors” (NCOMMS-21-34702) has been revised according to the insightful suggestions of the reviewers and editors and is hereby re-submitted for possible publication in the *Nature Communications*. A point-by-point reply to the reviewers’ comments follows.

Point-by-Point Reply to Reviewers’ Comments (the reviewers’ comments are in *italics*):

General comments from all three reviewers and editors:

Reviewers and editors mentioned that our original manuscript was descriptive and defective in issues such as language, sample size, multiple clustering schemes in analyzing high-dimensional data, evidence for epithelial origin of the tumours, and functional validation of the proposed cell-cell interactions. In this revised submission, we have performed additional experiments and revised the manuscript accordingly. Especially, we added additional data to strengthen the validity, robustness, and significance of our immuno-reclassification system. In addition, we would like the title of our manuscript to be revised to “A cell atlas-derived immuno-reclassification of human thymic epithelial tumors” in order to better emphasize the findings. We believe that our revised manuscript is obviously improved by the kindly help of reviewers and editors. Areas underlined is added new data or rephrased in the revised manuscript.

Reviewer #1 (Single cell analysis of TIL):

In this manuscript, Xin et al. conducted a single-cell study of thymic tumors based on CyTOF and single-cell RNA sequencing. A new classification of thymic tumors was proposed and multiple characteristics were discussed. While this study is interesting, critical concerns exist before publication.

Response: Thanks the reviewer for appreciating the quality of our work and for your thoughtful comments.

1. It is unclear how the new classification was derived. Since single-cell data including CyTOF and scRNA-seq were high-dimensional, it is expected that multiple clustering schemes exist. The validity, robustness, and significance of the current classification should be demonstrated. However, the current manuscript is mainly descriptive. Comparison with alternative clustering schemes including WHO classifications was absent, limiting the significance of this study.

Response: Thanks for the comment and helpful suggestion. As TETs are originated from thymic epithelial cells (TECs), which play a crucial role in T-cell development, we hypothesized that the immune landscape of tumor was impacted by the malignant transformation of TECs during the development of human TETs. As expected, an obvious difference was observed between the immune landscape (shown in t-SNE maps) of the tumor and normal thymus tissue derived from CyTOF data (revised Figure 1b and Supplementary Figure 1a). Moreover, a remarkable difference of cell composition among samples of human TETs was also observed (revised Supplementary Figure 1a). So we classified the tumor samples into three types according to the similarity of cell clusters shown in the t-SNE maps (revised Figure 1b and Supplementary Figure 1a). Through cell annotating, we found that the immature T cells and mature T cells are the core discrepancy

among samples (revised Figure 1h, j and Figure 2a, c). This is the origin of our new classification of human TETs.

According to the kindly suggestion of R#1 and editor, we performed multiple cluster schemes (revised Supplementary Figure 1b, c) and added some new data to further confirm the validity, robustness and significance of our classification in distinguishing the discrepancy of cellular composition among samples of human TETs compared with WHO classifications and Masaoka stage. As shown in revised Figure.1b-d and Supplementary Figure 1e-f, our reclassification is superior to both WHO classifications and Masaoka stage in distinguishing the discrepancy of cellular composition among samples, including CD3-CD4+CD8+ cells, B cells, DCs and granulocytes (revised Figure.1b, d, j, k, Supplementary Figure 1e, f and Supplementary Figure 3). Moreover, we found that compared with WHO classifications and Masaoka stage, our reclassification showed greater advantages in distinguishing the difference of cell composition associated with intrathymic T cell development among tumors (revised Figure.2c-g and Supplementary Figure 5). Taken together, we believe that these data further supported the validity, robustness, and significance of our reclassification in distinguishing the difference of cell composition among samples of human TETs.

2. Since CyTOF data were on the protein level while scRNA-seq data were on the RNA level, a comparison between CyTOF findings with the observations based on scRNA-seq should be explicitly presented regarding both cell proportion and molecular characteristics.

Response: Thanks for the helpful suggestion. We performed a comparison between CyTOF findings with the observations based on scRNA-seq as suggested by R#1. Except for a mild discordance of T cell and DN cell proportions, most findings of CyTOF were validated in scRNA-seq data (revised Supplementary Figure 6h, 7c, i and 13c).

3. The discussion on the impact of tumors on T cell development is interesting. However, the current analysis seems to be superficial. As extrinsic T cells could be recruited to the thymus because of the immunogenicity of thymic tumors, the observations of defects in T cell development may result from the variance in sample collection rather than the biological impacts of tumor on T cell development. The authors should add new analyses to evaluate such possibilities.

Response: Thanks for the kindly comment and suggestion. We agreed that extrinsic T cells may be recruited to the thymus due to the immunogenicity of thymic tumors. As shown in the picture below, the thymus tissue and thymic tumors in resected specimen could be identified easily by our researchers (as shown in photograph below). Furthermore, the gross specimen and histological type of samples we collected were reconfirmed by two experienced thoracic surgeons and two pathologists individually. Therefore, a correct sample collection in our study is guaranteed.

The gross specimen of human TETs and thymus and adipose tissue. We can notice that tumor have an obvious capsule and a fish-like texture. And the tumor is always solid so we can recognize it easily.

In order to investigate the potential impact of thymic tumor cells on T cell development, we performed additional co-culture experiment of purified thymic tumor cells and CD3+CD8+CD4+T cells *in vitro* (revised Supplementary Figure 12d-i). The results showed that tumor cells from type 1 and type 2 TETs did could directly promote the generation of SP cells from DP cells *in vitro*. Taken together, we believe that the change of T cell development in tumors of human TETs observed in our study was at least partially derived from the biological impacts of tumor cells.

4. The evidence supporting the epithelial origins is weak. A deeper analysis including lineage tracing based on mitochondrial mutations or RNA velocity should be added. Otherwise, independent validation experiments should be added.

Response: Thanks for the comment and helpful suggestion. According to the kindly suggestion of R#1 and editor, we performed additional RNA velocity analysis of thymic tumor cells and found that mcTEC-like tumor cells were the precursor of most tumor cell subsets (6 in 7 clusters) in type 1 and type 2 of human TETs, including KRT14+ mTEC-like cells and GNB3+ mTEC-like cells (revised Supplementary Figure 9c and 10b). The result of RNA velocity analysis could support our conclusion to some extent.

5. Related to 4, it is difficult to directly derive the conclusion that epithelial origins had important impacts on the deficiency of T cell development. Alternative factors including acquired characteristics of different tumors could also contribute to the distortion of T cell development. Evidence supporting the roles of epithelial origins should be added.

Response: Thanks for the comment and suggestion. It is indeed difficult to directly derive the conclusion that epithelial origins had important impacts on the deficiency of T cell development in our study. However, we believe that the additional RNA velocity analysis (revised Supplementary Figure 10b) and co-culture experiment (revised Supplementary Figure 12d-i) suggested by R#1 supports the link of epithelial origins and T cell developmental changes in type 1 and type 2 of human TETs.

Minor concerns:

6. The language should be proofread by a native English speaker.

Response: Thanks for the suggestion. We have revised our language accordingly.

7. The figures can be further improved for clarity.

Response: Thanks for the suggestion. We have re-organized our figures throughout the article to improve the clarity.

Reviewer #2 (Thymic T cell development):

An improved classification of thymic epithelial tumors based on their molecular and cellular characteristics is important towards better understanding of those tumors and their consequences in the immune system and towards better treatment of the disease. However, this study described only 18 cases (2 A, 5 AB, 1 B1, 4 B2, 2 B3, and 3 C) in total and showed single cell RNA sequencing results only from 6 patients. Such a small number of the subjects critically diminishes the strength of this study. The new classification of three types sounds provocative but describes no objective basis other than graphical impression of CyTOF data shown in Figure 1B. Indeed, the manuscript offers no evaluation of pretexting WHO classification or Masaoka stages (what is improved in their new classification? what was wrong with the conventional technology). In addition, the manuscript offers no detailed description regarding how tumor cells and infiltrating hematopoietic cells were separated from neighboring cells, hampering to distinguish between the cells within the tumor and the cells proximal but outside the tumor. Therefore, the data shown in this manuscript are far from providing convincing or novel conclusion in terms of the re-classification and characterization of thymic epithelial tumors and associated thymocytes.

Response: Thanks the reviewer for appreciating the significance of our work and for your thoughtful critiques.

In our revised manuscript, a total of 42 TETs samples were detected in multiple research approaches, including CyTOF detection, multiple clustering schemes, single cell RNA sequencing, *in vitro* co-culture experiment, FCM detection and IF validation. Besides, we conducted quantitative statistics to validate the objective basis of our new classification (revised Figure 1c, d). We have compared the validity of our classification in distinguishing the difference of immune cell composition among tumors with WHO classification and Masaoka stages, and the results support the conclusion that our reclassification is superior in identifying the discrepancy of immune cell composition among tumors (revised Figure 1b, 2c-f and Supplementary Figure 3a-d, 5a, b). It is worth noting that, using another cohort of 119 patients from TCGA, we further demonstrated that our classification also showed advantages in prognostic prediction than both WHO classification and Masaoka stages.

Regarding to the sample collection, as response to R#1 above, we provided the picture of the gross specimen resected by an experienced thoracic surgeon and follow a rigorous procedure to confirm the histological type by two pathologists individually to ensure sampling quality and accuracy.

Taken together, according to the suggestion of reviewers and editors, we have increased our sample size, validated the superiority of our immune-classification, and added some new data to illuminate the underling mechanisms of impact of tumor cells on intra-tumoral T cell development. We believe that our revised manuscript was markedly improved.

Reviewer #3 (TECs/Thymic T cell development):

The manuscript by Xin et al is generally interesting, with lots of data and potentially important clinical implications. However, its major weakness is that it is purely descriptive and somewhat speculative.

Response: Thanks the reviewer for appreciating the workload, significance of our work and for your thoughtful comments. In this revised manuscript, we have enlarged the sample size, performed multiple clustering schemes, and demonstrated that our new classification is superior in identifying the discrepancy of immune landscape of human TETs among individuals compared to WHO classification or Masaoka stages. Regarding to the epithelial origin of human TETs, we performed RNA velocity analysis as recommended by R#1, and the results supported our conclusion. Furthermore, we performed additional co-culture experiment to validate the impact of tumor cells on T cell development, and found that tumor cells isolated from both type1 and type2 TETs did could directly induce SP generation from DP cells *in vitro*. Taken together, thanks to the kindly comments and suggestions of all Reviewers and Editors, we believe that our revised manuscript was significantly improved.

The authors largely rely on expression data to make claims regarding cell-cell interactions, cell recruitment or cells' effect without providing any further proof/validations for these claims.

Response: Thanks the reviewer for your thoughtful comments. We performed additional experiments to validate the tumor cell-mediated recruitment of CD8+TRMs, and demonstrated that tumor-infiltrating CD8+TRMs expressed higher level of CXCR3 by FCM analysis, which was consistent with the findings in mRNA level (revised Figure 6 e and f). Furthermore, we also validated the tumor cell-mediated recruitment of CD8+TRMs by a co-culture experiment *in vitro*, and demonstrated that tumor cells isolated from type3 TETs did could directly recruit CD8+TRMs (as shown in picture below). In order to investigate the potential immune cell recruitment effect of CD8+TRMs, we detect the expression level of CXCL13, a chemokine with a crucial role in multiple immune cells recruitment, such as CD4+TFH cells, CXCR5+CD8+ T cells and B cells. Consistent with the findings in mRNA level, we demonstrated that tumor-infiltrating CD8+TRMs of type3 TETs did could produce higher level of CXCL13 than any other T cells (revised Figure 6 m and n). These revised data combined with the findings of co-localization of CD8+TRMs with epithelial cells, DCs and B cells in situ by IF staining, jointly supported our conclusion about cell recruitment related to tumor-infiltrating CD8+TRMs.

As shown in left, we performed a recruiting experiment *in vitro*, and demonstrated a direct recruitment of CD8+TRMs by type 3 tumor cells of human TETs. (The digit in scatter plot were cell numbers of recruited CD8+TRMs; n=1)

The other major weak point of the manuscript is the clarity of the text and of data presentation. The text requires serious language edits with a professional English editor. Some parts of the text are hard to understand. Moreover, the figure legends lack a lot of critical information. For example, the authors do not mention if a heatmap is normalized per row/column/total matrix. This is

especially problematic in 2B where it seems that it's definitely not normalized per gene, if it's normalized per cluster only (within each cluster) it's completely meaningless (different expression level of different genes within the same cells doesn't mean anything, some genes need to be expressed highly to have an effect, while others only need a few molecules to work..). I would hope that it's normalized for the entire matrix, which is a strange choice but at least it would make sense. I would suggest they add this information, and normalize per gene if they haven't done it.

Response: Thanks the reviewer for your precise critiques and helpful suggestions. We have re-organized our text and figures in our revised manuscript to improve the clarity of the text and data presentation. Regarding the language, it has been revised by a professional English editor from SPRINGER NATURE Author Service. Many thanks R#3 for pointing out our deficiency in figure legends and heatmap normalization, we have added the normalized information in the legend of each heatmap. As for Fig 2b, we had normalized total matrix previously, and the normalized data are as below.

#	A	B	C	D	E	F	G	H	I	J	K	L	M	N	O	P	Q	R	S	T	U	V	W	X	Y
1	Naive CD4 ⁺ T cell	CD45	CD3	CD4	CD8	CD45RO	CD103	CD59	FoxP3	CTLA4	PD_1	CD101	CD39	CD38	CD49a	CD137	CD28	K67	Granzyme	Tbet	BCL6	ECMES	RORγt	GATA3	CD27
2	Memory CD4 ⁺ T cell	0.914097	0.917686	0.835771	0	0.257788	0	0.209927	0.127923	0.095427	0	0	0	0.435402	0	0.594032	0.222254	0	0	0	0	0	0	0.0391	0.754755
3	CD4 ⁺ Treg	0.804853	0.791661	0.850956	0	1	0.825054	0.660182	0.097414	0.362624	0.573504	0.03754	0.171712	0.348382	0.131603	0	0.623601	0.335325	0	0.030228	0	0.018887	0.006322	0.006552	0.124107
4	Naive CD8 ⁺ T cell	0.998471	0.846433	0.815825	0.001985	1	0	0.638815	0.459759	0.588074	0.423439	0	1	0.476196	0	0.063157	0.855122	0.40135	0	0.126948	0	0.059251	0.0654	0.129278	0.457114
5	Memory CD8 ⁺ T cell	0.97216	0.798735	0	0.839863	0.196922	0	0.375581	0.152473	0.103745	0	0.051144	0	0.200949	0	0	0.531212	0.278201	0	0	0	0.048248	0.045426	0.072304	0.801489
6	DN T	0.994291	0.790799	0.004414	0.840763	0.810503	0	0.606675	0.113574	0.132198	0.143002	0.002442	0	0.78282	0.045174	0	0.545394	0.26652	0	0	0	0.042871	0.009987	0.059132	0.661268
7	DP T	0.834427	0.769931	0	0.85238	1	1	0.91225	0.143145	0.387777	0.675867	0.070822	0.678795	0.606022	0.745008	0.01554	0.156291	0.352411	0.095777	0.05649	0	0.0605	0.01129	0.027069	0.214999
8	DP T	0.976936	0.821667	0.950065	0	0.946732	0	0.342567	0.197359	0.170254	0.716975	0	0	0.943006	0	0	0.728753	0.248215	0	0	0.069946	0.021736	0.126746	0.491106	
9	DP T	0.73118	0.553954	0.642124	0.606678	1	0.039883	0.12644	0.118077	0.036628	0	0	0	0	1	0	0.499498	0.505114	0	0	0.444557	0	0	0.230033	

Another example is that they don't say what are the squares in 2B,D.

Response: Thanks the reviewer and we have corrected it accordingly.

While the classification of type1 and 3 are convincing (Fig 1B), the classification of type 2 seems less convincing ... 04 seems more like normal, and 05 looks completely different from 10 and 11, which themselves look more like type1. Thus, the overall classification of the tumors as proposed by the authors is questionable. Most common tumor seems to be type 1, then type 3, there seem to be some other less common tumors like 04 and 05, which in my opinion do not fit into the 3 model classification.

Response: Thanks the reviewer for your critiques. We do agree that the classification of type2 TETs was less convincing than type1 and type3 in our previous manuscript. In order to further validate the validity, robustness, and significance of our classification, we enlarged the sample size of CyTOF detection and performed multiple clustering schemes in our revised manuscript. We noticed that the difference of immune cell composition among type1, type2 and type3 TETs was further validated in additional samples. For instance, as shown in revised Figure 1h-k, the proportion of CD3-CD4+CD8+ cells, CD3+CD4+CD8+ cells, B cells DCs and granulocytes in tumor were different among three types (revised Figure 1 h-k). It is worth noting that, after enlarging the sample size, the difference of immune cell composition among type1, type2 and type3 TETs were becoming more obvious (revised Supplementary Figure 1a). Through deep cell annotating, we found that the T cell subsets involved in intra-thymic T cell development were more distinct among TETs types, especially among type2 and typ1, and among type2 and typ3 (revised Figure 2 c-g). These results suggest that our tripartite classification could preferably reflect the discrepancy of immune landscape, especially cell composition relate to intra-thymic T cell development, among tumors of human TETs.

Moreover, when compared with normal thymus the CD3-CD4+CD8+ cells and CD3+CD4+CD8+ cells which involved in early intra-thymic T cell development, were significantly increased in type1 TETs (revised Figure 1 j, k and revised Figure 2 c-g), and T cell subsets involved in later stage of intra-thymic T cell development such as CD4+ T cells, CD8+T cells, Naïve CD4+ cells, Memory

CD4+ T cells and CD4+ Treg cells, were significantly decreased in type1 TETs (revised Figure 2 c-g). In contrast, the cell subsets involved in early intra-thymic T cell development were decreased in type3 TETs compared with normal thymus (revised Figure 1 j, k and revised Figure 2 c-g) and T cell subsets involved in later stage of intra-thymic T cell development such as CD4+ T cells, CD8+T cells, Naïve CD4+ cells, Memory CD4+ T cells and CD4+ Treg cells, were significantly increased in type3 TETs (revised Figure 2 c-g). However, when compared with normal thymus the cell subsets involved in both early intra-thymic T cell development and later intra-thymic T cell development were not significantly varied in type2 TETs (revised Figure 1j, k and revised Figure 2 c-g). These results suggested our classification could also reflect the changes of immune landscape of tumors of human TETs from normal thymus.

Furthermore, through performing multiple clustering schemes we demonstrated that our classification could distinguish the difference of immune cell composition among samples, such as CD3-CD4+CD8+ cell, B cell, DCs and granulocytes, better than both WHO classifications and Masaoka stage (revised Figure.1h-k and Supplementary Figure 3). Moreover, we found that the difference of cell composition related to intra-thymic T cell development among tumors was also better distinguished by our classification (revised Figure.2c-f and Supplementary Figure 5).

Taken together, through enlarging sample size of CyTOF detection and performing multiple clustering schemes, our classification was further improved in validity, robustness, and significance. We believe that these additional works in our revised manuscript has make our classification of human TETs, including type2, more convincing.

Generally, it is unclear to me from the methods how the authors decided to group the samples into the 3 types.

Response: Thanks the reviewer for your comment. Briefly, at first we grouped the samples of TETs according to WHO classifications and Masaoka stage. However, we found the discrepancy of immune cell composition among groups was not distinct, such as especially among A, AB and B of WHO classifications, and I and II stage of Masaoka stages. Therefore, we decided to regroup the samples according to the similarity of t-SNE maps (revised Supplementary Fig. 1a). We have added the t-SNE maps of samples grouped according to WHO classifications and Masaoka stage in our revised manuscript (revised Figure 1b).

Moreover, since the tumors are epithelial tumors, it would make more sense to present the classification based on epithelial cells from the very beginning.

Response: Thanks the reviewer for your comment. We appreciated the opinion of presenting the classification based on epithelial cells from the very beginning. However, it is not consistent with our research process. At the beginning of our study, we aimed to uncover the immune landscape of human TETs, and then derived our classification based on the discrepancy of immune cell composition among tumors. After that, in order to validate our classification and find underlying mechanism, we choose to perform single cell RNA sequencing of typical samples according to our classification confirmed by CyTOF detection. And then, epithelial cells in each type of tumor were identified. So, we prefer to present our data according to our research process. Besides, through cell annotating we found CD45+ immune cells, but not EPCAM+ tumor cells, constituted the most dominant cell subset in TET tumors (revised Figure. 1g and Supplementary Figure. 1g). Moreover, the quantity of epithelial cells less varied among tumor types than immune cells (revised Figure. 1b, g and Supplementary Figure. 1h).

Correspondingly a much better characterization of the epithelial subsets should be provided and compared with TEC subsets in healthy thymus. For instance GNB3 is a marker of tuft cells – Does this tumor express additional tuft cell markers? The characterization of Krt14+ cells is very non-informative – more markers are needed to better characterize this subset.

Response: Thanks the reviewer for your comment. Additional molecular characteristics of tumor epithelial cells were added in the revised manuscript and results. We found although tumor epithelial cells retained some molecular characteristics of normal thymic epithelium, the expressions of other genes were changed. For example, GNB3+ mTEC-like cells expressed high levels of *OVOL3*, *PLCB2*, *TRPM5*, but low level of *PLCG2*, *CHAT*, *DCLK1*, all of which are the marker genes of tuft-like mTEC cell in normal thymus^{1, 2, 3} (revised Supplementary Fig. 9e). Similarly, *KRT14*+ mTEC-like cells expressed high level of *CLU*, *FN1*, *IGFBP5*, but not of *ITGA6*, *KRT17*, *SOX4*, all of which are the marker genes of *KRT14*+mTEC(I) cell in normal thymus^{1, 3}(revised Supplementary Fig. 9f). In addition, *KRT14* was used to define mTEC(I) cells in Jong-Eun Park et al. 's study¹. Therefore, we believe that *KRT14* could define this tumor epithelial subset in TETs, although it may not be specific for epithelial cells in other tissues.

More phenotypic data should be presented for the patients with the individual tumor types. The data suggest that patients who developed type 3 tumor, had poorer survival. Is information about development of autoimmune diseases (e.g. myasthenia gravis, which occurs in large fraction of thymoma patients) available? Could this be correlated with the expression of acetyl choline receptor (CHRNA) in the individual types

What about other symptoms, such as immunodeficiency or immune dysregulation? Would be interesting to correlate it the type of the tumor with the specific phenotype.

Response: Thanks the reviewer for your comment. We reviewed the autoimmune diseases and other symptoms of patients in our study. Unfortunately, except for myasthenia gravis, other autoimmune diseases and symptoms of patients were not acquired. However, we found the myasthenia gravis were not presented in the patients of type3, but presented in the patients of type1 and type2 (revised Supplementary Figure 11 b). It was interesting that the incidence rate of myasthenia gravis was mildly higher in the patients of type1 than type2 (revised Supplementary Figure 11 b). As for *CHRNA1*, we did not find an obvious correlation with the occurrence of MG (revised Supplementary Fig. 11c). The correlation of autoimmune diseases and symptoms and our classification of TETs needs further investigation in the future.

-Fig. 2B- how can the DP T cells be CD45RO+?

Response: Thanks the reviewer for your comment. We validated the expression of CD45RO on DP T cells by additional FCM analysis. As shown in data following, we confirmed the expression of CD45RO on DP T cells by a rigorous FMO control. We do not know why DP T cells in tumor of human TETs expressed CD45RO yet, it is still an open question.

Detection of CD45RO by FCM and FMO control

"To solve this divergence, we further annotated DP cells into proliferating (P) and quiescent (Q) subsets according to the expression level of MKI67 and CDK1 as previous study 21 (Supplementary Fig. S4C and S4D). Consistently, the marker genes of CD3- DP cells, which were obtained from the protein marker-based sorting strategy in another study 23, were obviously highly expressed in DP (P) cells (Supplementary Fig. S4E). These results suggested a close match between the DP (P) cells defined by scRNA-seq and CD3- DP cells detected by CyTOF in our study." I didn't really understand authors' explanation about the discrepancy between the RNA expression and CyTOF data for CD3 in DP cells- how does proliferation explain why CD3-expressing DP cells don't bind the antibody for CD3?

Response: Thank the reviewer for your comment. We are sorry that our previous description was not accurate enough, which caused the reviewer's confusion. We have revised it in the manuscript (revised manuscript line 180-187). In addition, corresponding experiments are added to prove that DP (P) cells defined by scRNA-seq were closely matched to CD3- DP cells detected by CyTOF (revised Supplementary Figure 7g, h). These data suggest the expression levels of CD3 gene and CD3 protein in CD3- DP cells were inconsistently, which explained why we didn't find CD3- DP cells from the scRNA-seq results.

6l – The authors show the cells are in proximity to epithelial cells but this is very general, are they close to the CHI3L1 cells in particular?

Response: Thanks the reviewer for your comment. We validated the co-localization of CD8+ TRMs and CHI3L1+ cells of type3 TETs in situ by additional IF analysis. As shown in the revised data (revised Supplementary Figure 13 d), we confirmed the co-localization of CD8+ TRMs and CHI3L1+ cells in tissue of type3 tumor using multiple fluorescence staining of CD8, CD103 and CHI3L1.

-in the 6th chapter, could they provide an actual proof, any additional validation other than expression data for their claims? For example, the activation of CD8 Trm cells or their recruitment of other immune cells? Ex vivo even... Or even just the actual protein overexpression of the ligands raised in 6G... Or for the ligand-receptor interactions between the CD8 Trm cells to CHI3L1 epithelial cells...I think any additional validation will strengthen their paper, this chapter feels a bit speculative as it is.

Response: Thanks the reviewer for your comment and helpful suggestions. We validated the expression levels of CXCR3, IFN- γ , CD39 and CXCL13 of CD8+ TRMs in type3 TETs by additional FCM analysis. As showed in the revised data (revised Figure 6 e-f, i-f, and m-n), higher expression levels of CXCR3, IFN- γ , CD39 and CXCL13 on CD8+ TRMs were confirmed. Furthermore, as shown in picture below, we also validate the tumor cell-mediated recruitment of CD8+TRMs by a co-culture experiment *in vitro*, and demonstrated that tumor cells isolated from type3 TETs could directly recruit CD8+TRMs.

As shown in left, we performed recruiting experiment *in vitro*, and demonstrated a direct recruitment of CD8+TRMs by type 3 tumor cells of human TETs. (The digit in scatter plot were cell numbers of recruited CD8+TRMs; n=1)

References:

1. Park JE, *et al.* A cell atlas of human thymic development defines T cell repertoire formation. *Science* **367**, (2020).
2. Miller CN, *et al.* Thymic tuft cells promote an IL-4-enriched medulla and shape thymocyte development. *Nature* **559**, 627-631 (2018).
3. Bornstein C, *et al.* Single-cell mapping of the thymic stroma identifies IL-25-producing tuft epithelial cells. *Nature* **559**, 622-626 (2018).

REVIEWER COMMENTS

Reviewer #1 (Remarks to the Author):

The authors have addressed most of my concerns. My remaining concern is the robustness of the newly proposed classification because the new classification is built on clustering in high-dimensional data. It is better to derive a simplified set of marker genes to reproduce the classification.

Reviewer #2 (Remarks to the Author):

Revised manuscript still provides only a small set of data (26 cases for CyTOF and 6 cases of scRNAseq), so that the study remains inconclusive especially in terms of the advantage over preexisting WHO classification and Masaoka stages.

Reviewer #3 (Remarks to the Author):

The revised manuscript has been significantly improved. I also commend the authors for making a serious effort to dramatically improve the language of the paper.

The data are much more convincing now and suggest that there are at least three major types of thymic tumors, which could be distinguished by the markers suggested by the authors. In addition, the study could serve as a valuable resource for the scientific community. Therefore, in my opinion the revised study is generally suitable for publication in Nat Comm.

I however have a few minor comments and suggestions

I believe that the data related to TEC subsets in Fig 5 and corresponding supplementary figures could be presented in a more informative way

- It would be very interesting and informative to show the projection of several key TEC genes in the UMAP graphs presented in Fig5e and show their representation in the three main tumor subsets. Alternatively similar graphs as in 5c should be presented for more TEC marker genes (Aire, Ccl25, Chi3l, Myog, chga, Gnb3, Fezf2, chrna1, etc..)

- Fig 5f – the authors should present at least 3-5 most specific genes that characterize the individual population. Some populations are highlighted only by a single gene. Eg. What other genes are highly enriched in the Myog, Chga, etc, clusters?

- Fig.5i-j-k-l don't seem essential for the main figure, and could be moved to supplement

- The authors claim that there is no correlation between MG and the type of tumor and reference Suppl Fig 11c to support this claim. However the this figure does not show any correlation analysis, only expression of Chrna1 in different TEC subsets. Interestingly, T3 is the only tumor that did not result in MG and it is the only tumor that showed detectable expression of Chrna1 in the Myog TEC subset. These results indicate that loss of the Chrna1 expression MyoG TEC in T1 and T2 patients may in fact result in MG development in some of these patients.

The authors should revise accordingly

June 17, 2022

Editorial Office
Nature Communications

The enclosed manuscript entitled “A cell atlas-derived immuno-reclassification of human thymic epithelial tumors” (NCOMMS-21-34702A) has been revised according to the insightful suggestions of the reviewers and editors and is hereby re-submitted for possible publication in the *Nature Communications*. A point-by-point reply to the reviewers’ comments follows.

Point-by-Point Reply to Reviewers’ Comments (the reviewers’ comments are in *italics*):

Areas underlined is added new data or rephrased in the revised manuscript.

Reviewer #1 (Remarks to the Author):

The authors have addressed most of my concerns. My remaining concern is the robustness of the newly proposed classification because the new classification is built on clustering in high-dimensional data. It is better to derive a simplified set of marker genes to reproduce the classification.

Response: Thanks the reviewer for your comments. By using marker genes of tumor cells in type 1 and type 3 TETs (GNB3 and CHI3L1 for type 1 and type 3 respectively), we have reproduced our classification in another cohort of 119 TET patients from the TCGA (Fig. 7**a-c** and Supplementary Fig. 15**a**) and demonstrated that our classification had advantages in prognostic prediction compared with both Masaoka staging system and WHO classification (Fig. 7**d** and Supplementary Fig. 15**b, c**). Clinicians could reproduce the classification easily based on the expression levels of GNB3 and CHI3L1 through genetic testing, immunohistochemistry or immunofluorescence. Of course, we also believe that further clinical study is still needed to validate the value of our classification in prognostic prediction of human TETs.

Figure 7

Supplementary Figure 15

a

b

c

Reviewer #2 (Remarks to the Author):

Revised manuscript still provides only a small set of data (26 cases for CyTOF and 6 cases of scRNAseq), so that the study remains inconclusive especially in terms of the advantage over preexisting WHO classification and Masaoka stages.

Response: Thanks the reviewer for your comments. As far as we know, our study was the first study using CyTOF or scRNAseq to investigate the immune landscape of human TETs. In addition, using the marker genes of tumor cells identified in type 1 and type 3 TETs, we validated our classification in another cohort of 119 TET patients from the TCGA (Fig. 7a-c and Supplementary Fig. 15a) and showed that our classification had advantages in prognostic prediction compared with both the Masaoka staging system and WHO classification (Fig. 7d and Supplementary Fig. 15b, c). Although the value of our classification is still needed to be further validated in clinic, we believe that our study had provided valuable information in the research field of human TETs classification. Meanwhile, we also added a discussion of relevant content in the manuscript (line 410-413, “As expected, our classification of TETs showed advantages in prognostic prediction compared with the WHO classification and Masaoka staging system, although further clinical study was still needed to validate the value of our classification.”).

Reviewer #3 (Remarks to the Author):

The revised manuscript has been significantly improved. I also command the authors for making a serious effort to dramatically improve the language of the paper.

Response: Thanks the reviewer for your comments.

The data are much more convincing now and suggest that there are at least three major types of thymic tumors, which could be distinguished by the markers suggested by the authors. In addition, the study could serve as a valuable resource for the scientific community.

Therefore, in my opinion the revised study is generally suitable for publication in Nat Comm.

Response: Thanks the reviewer for your thoughtful comments.

I however have a few minor comments and suggestions

I believe that the data related to TEC subsets in Fig 5 and corresponding supplementary figures could be presented in a more informative way

- It would be very interesting and informative to show the projection of several key TEC genes in the UMAP graphs presented in Fig5e and show their representation in the three main tumor subsets. Alternatively similar graphs as in 5c should be presented for more TEC marker genes (Aire, Ccl25, Chi3l, Myog, chga, Gnb3, Fezf2, chrna1, etc..)

Response: Thanks the reviewer for your helpful suggestions. We have added relevant content in Fig 5g, Supplementary Fig. 10, and Supplementary Fig. 12a, e accordingly.

- Fig 5f – the authors should present at least 3-5 most specific genes that characterize the individual population. Some populations are highlighted only by a single gene. Eg. What other genes are highly enriched in the Myog, Chga, etc, clusters?

Response: Thanks the reviewer for your helpful suggestions. We have revised Fig5f accordingly.

- Fig.5i-j-k-l don't seem essential for the main figure, and could be moved to supplement

Response: Thanks the reviewer for your helpful suggestions. We have moved Fig.5i-j-k-l to Supplementary Fig. 13.

- The authors claim that there is no correlation between MG and the type of tumor and reference Suppl Fig 11c to support this claim. However the this figure does not show any correlation analysis, only expression of Chrna1 in different TEC subsets. Interestingly, T3 is the only tumor that did not result in MG and it is the only tumor that showed detectable expression of Chrna1 in the Myog TEC subset. These results indicate that loss of the Chrna1 expression MyoG TEC in T1 and T2 patients may in fact result in MG development in some of these patients. The authors should revise accordingly

Response: Thanks the reviewer for your helpful suggestions. We have added the expression level of *CHRNA1* gene in epithelial cells among different types of TETs in Suppl Fig 12e and rephrased the text about Suppl Fig 12d and e into “It is interesting that expression of *CHRNA1* was only detected in MYOG+ TEC-like cells of type 3 TETs, which indicated that the loss of *CHRNA1*+ MYOG+ TEC-like cells may involve in the MG development in type 1 and type 2 TETs” (line **284-288**).

REVIEWERS' COMMENTS

Reviewer #1 (Remarks to the Author):

The authors have addressed all my concerns.

Reviewer #2 (Remarks to the Author):

This manuscript challenges conventional classification of thymic epithelial tumors, including the WHO classification and the Masaoka system. Those conventional classifications have been widely used based on the studies of exceeding numbers of patients (for example, more than 6,000 case in Marx, et al. J Thorac Oncol. 2015; 1,320 cases in Kondo, et al. Ann Thorac Surg 2003; 537 cases in de Jong, et al. Eur J Cancer 2008; 1,930 cases in Wu, et al. J Thorac Dis. 2016) and have been continuously evaluated and revised (for example, analyzing 250 cases in Hadoux, et al. Bull Cancer. 2012; 286 cases in Okumura, et al. Jpn J Thorac Cardiovasc Sug. 2002; Marx, et al. J Thorac Oncol. 2014). In contrast, this manuscript based on scRNAseq analysis of only 6 cases and CyTOF analysis of additional 20 cases (summarized in Supplementary Table 2) describes the conclusions that the WHO classification and Masaoka stage are "unsatisfactory" and that the authors "successfully reclassified" thymic epithelial tumors. It is sad and rather painful to find such scientifically poor and arrogant statements in this manuscript based on its limited analysis.

Reviewer #3 (Remarks to the Author):

The authors have addressed my comments and I am happy to recommend the manuscript for publication

August 14, 2022

Editorial Office
Nature Communications

The enclosed manuscript entitled “The immune landscape of human thymic epithelial tumors” (NCOMMS-21-34702B) has been revised according to the insightful suggestions of the reviewers and editors and is hereby re-submitted for possible publication in the *Nature Communications*. A point-by-point reply to the reviewers’ comments follows.

Point-by-Point Reply to Reviewers’ Comments (the reviewers’ comments are in *italics*):

Areas with ‘Track Changes’ feature is rephrased in the revised manuscript.

Reviewer #1 (Remarks to the Author):

The authors have addressed all my concerns.

Response: Thanks Reviewer #1 for your comment.

Reviewer #2 (Remarks to the Author):

This manuscript challenges conventional classification of thymic epithelial tumors, including the WHO classification and the Masaoka system.

Response: Thanks Reviewer #2 for your comment. We apologize for the inappropriate statement of the superiority of our classification in previous revision. We have toned down the classification claims as the editor’s suggestion and revised the manuscript accordingly.

Those conventional classifications have been widely used based on the studies of exceeding numbers of patients (for example, more than 6,000 case in Marx, et al. J Thorac Oncol. 2015; 1,320 cased in Kondo, et al. Ann Thorac Surg 2003; 537 cases in de Jong, et al. Eur J Cancer 2008; 1,930 cases in Wu, et al. J Thorac Dis. 2016) and have been continuously evaluated and revised (for example, analyzing 250 cases in Hadoux, et al. Bull Cancer. 2012; 286 cases in Okumura, et al. Jpn J Thorac Cardiovasc Sug. 2002; Marx, et al. J Thorac Oncol. 2014).

Response: Thanks Reviewer #2 for your comment. We deeply respect the contribution of clinical studies which derived the guidelines of conventional classifications of thymic epithelial tumors. However, it is worthy to note that our study is a translational research which trying to understand the biology of human thymic epithelial tumors in the perspective of immune tumor microenvironment.

In contrast, this manuscript based on scRNAseq analysis of only 6 cases and CyTOF analysis of additional 20 cases (summarized in Supplementary Table 2) describes the conclusions that the

WHO classification and Masaoka stage are “unsatisfactory” and that the authors “successfully reclassified” thymic epithelial tumors.

Response: Thanks Reviewer #2 for your comment. We agree that the sample size of our study is not enough for clinical studies mentioned by Reviewer #2. However, our study is a translational study which trying to understand the characteristic and formation of immune microenvironment of human TETs. Besides, we had increased the sample size as required by Reviewer #2 in the 1st revision, and our major findings were validated. Hence, we believe that further persist in using the standard of clinical study to judge our sample size may not necessary.

On the other hand, thanks to the criticism of Reviewer #2 and kindly suggestion of Editors, we realized that overstate the superiority of our classification is not appropriated. Therefore, we have revised the manuscripts throughout to tone down our classification claims. The inappropriate statement mentioned by Reviewer #2 was revised accordingly.

It is sad and rather painful to find such scientifically poor and arrogant statements in this manuscript based on its limited analysis.

Response: Thanks the Reviewer #2 for your criticism. We sincerely apologize for the inappropriate statements which make you uncomfortable. We have toned down our classification claims and revised the manuscript accordingly.

Reviewer #3 (Remarks to the Author):

The authors have addressed my comments and I am happy to recommend the manuscript for publication

Response: Thanks Reviewer #3 for your kindly recommendation.